# Development of a semi-Lagrangian advection scheme for the NEMO ocean model (3.1)

Christopher Subich[1], Pierre Pellerin[1], Gregory Smith[1], and Frederic Dupont[1]

[1]Environment and Climate Change Canada, 2121 route Transcanadienne ouest, Dorval Québec, Canada H9P 1J3

**Correspondence:** Christopher Subich (Christopher.Subich@canada.ca)

**Abstract.** As resolutions of ocean circulation models increase, the advective Courant number – the ratio between the distance travelled by a fluid parcel in one timestep and the grid size – becomes the most stringent factor limiting model timesteps. Some atmospheric models have escaped this limit by using an implicit or semi-implicit semi-Lagrangian formulation of advection, which calculates materially-conserved fluid properties along trajectories which follow the fluid motion and end at prescribed grid-points. Unfortunately, this formulation is not straightforward in ocean contexts, where the irregular, interior boundaries imposed by the shore and bottom orography are incompatible with traditional trajectory calculations.

This work describes the adaptation of the semi-Lagrangian method as an advection module for an operational ocean model. We solve the difficulties of the ocean's internal boundaries by calculating parcel trajectories using a time-exponential formulation, which ensures that all parcel trajectories remain inside the ocean domain despite strong accelerations near the boundary. Additionally, we derive this method in a way that is compatible with the leapfrog timestepping scheme used in the NEMO-OPA (Nucleus for European Modelling of the Ocean, Océan Parallélisé) ocean model, and we present simulation results for a simplified test-case of flow past a model island and for 10-year free runs of the global ocean on the quarter-degree ORCA025 grid.

## 1 Introduction

Recent work by Smith et al. (2018) has shown that over the medium term (up to seven days), a coupled forecasting system involving ocean, ice, and atmospheric models can significantly improve forecasting skill over forecasts that persist initial ocean and ice conditions over the atmospheric forecast period. While this is an exciting development for the future of numerical weather prediction, coupling adds a new dimension to the computational cost. Developing a deployable forecast system, especially with regional or ensemble components, requires exploiting every reasonable opportunity for optimization. One straightforward optimization is to maximize the admissible timestep of the ocean component, and we intend to improve the ocean timestep limit by implementing a semi-Lagrangian advection module into the popular NEMO-OPA (Nucleus for European Modelling of the Ocean, Océan Parallélisé; Madec (2008), version 3.1) model, used in this coupled system. This module is

intended as a drop-in replacement for the model's other advection modules, and in particular it does not interfere with NEMO's time-stepping algorithm (leapfrog).

## 1.1 Timestep constraints in the ocean

A numerical model with an explicit time-marching scheme must generally limit its timestep to satisfy a Courant-Friedrichs-Lwey (CFL) condition: information must not propagate more than a discretization-defined maximum number of cells in a single step, leading to a maximum stable Courant number. For systems such as the Euler equations (for the atmosphere) or hydrostatic equations (as implemented by NEMO-OPA), the information propagation speeds are controlled by the admissible wave modes of the systems, which become characteristic curves.

In the atmosphere, the most restrictive wave mode is that corresponding to sound waves. These waves are fast compared to atmospheric motions, and in response atmospheric models generally treat sound waves either implicitly or through sub-cycling, especially in the most restrictive vertical direction. The second most stringent restriction comes from simple advection by winds in the upper atmosphere. At the Canadian Meteorological Centre, the atmospheric forecasting system (and atmospheric component of the coupled forecasting system) uses the GEM (Geophysical Environmental Multiscale; Girard et al. (2014)) model, which addresses this timestep restriction through a semi-Lagrangian treatment of advection (Robert, 1982).

In the ocean, the Boussinesq assumption eliminates sound waves, but the model is left with the problem of surface gravity waves. Here, NEMO-OPA takes a similar approach to that used by atmospheric models for sound waves, by either treating the surface pressure gradient in a time-implicit manner (with a linearized free surface, used in this work), or by sub-cycling. The ocean lacks any direct equivalent to the atmosphere's strong upper-air winds, and so advection by the background velocity and internal gravity wave modes compete as the next most limiting factor for the maximum stable timestep. Lemarié et al. (2015) finds that the Courant number associated with vertical advection is more limiting than that associated with internal (baroclininc) gravity waves at resolutions of $\frac{1}{2}^{\circ}$, and the Courant number associated with horizontal advection catches up with that of gravity waves at resolutions of $\frac{1}{4}^{\circ}$ and finer.

### Grid stretching

In order to cover the entire ocean in a single, continuous domain, global NEMO-OPA model configurations typically use grids based on the ORCA "tripolar" grid (Madec and Imbard, 1996; Murray, 1996). This grid is defined in the northern hemisphere by an elliptical coordinate system, where the latitude-like coordinate is defined by ellipses with a shared pair of foci and the longitude-like coordinate is defined by the hyperbolas orthogonal to these ellipses. These coordinates match continuously at the equator to lines of latitude and longitude in a Mercator projection. By placing the foci of the ellipses on land, the grid contains no singularities in the ocean domain.

Unfortunately, this placement causes an abundance of small grid cells in the north polar region, especially in the Canadian Arctic Archipelago. Figure 1 depicts this situation at a nominal $\frac{1}{4}^{\circ}$ resolution: the grid point spacing of 25-30km near the equator falls to 3-4km in the archipelago. The areas in figure 1 with the narrowest grid spacing are also shallow seas, with depths of 200m or less and non-tidal currents of $15$–$30\mathrm{cm/s}$. This grid stretching is of particular concern when adapted

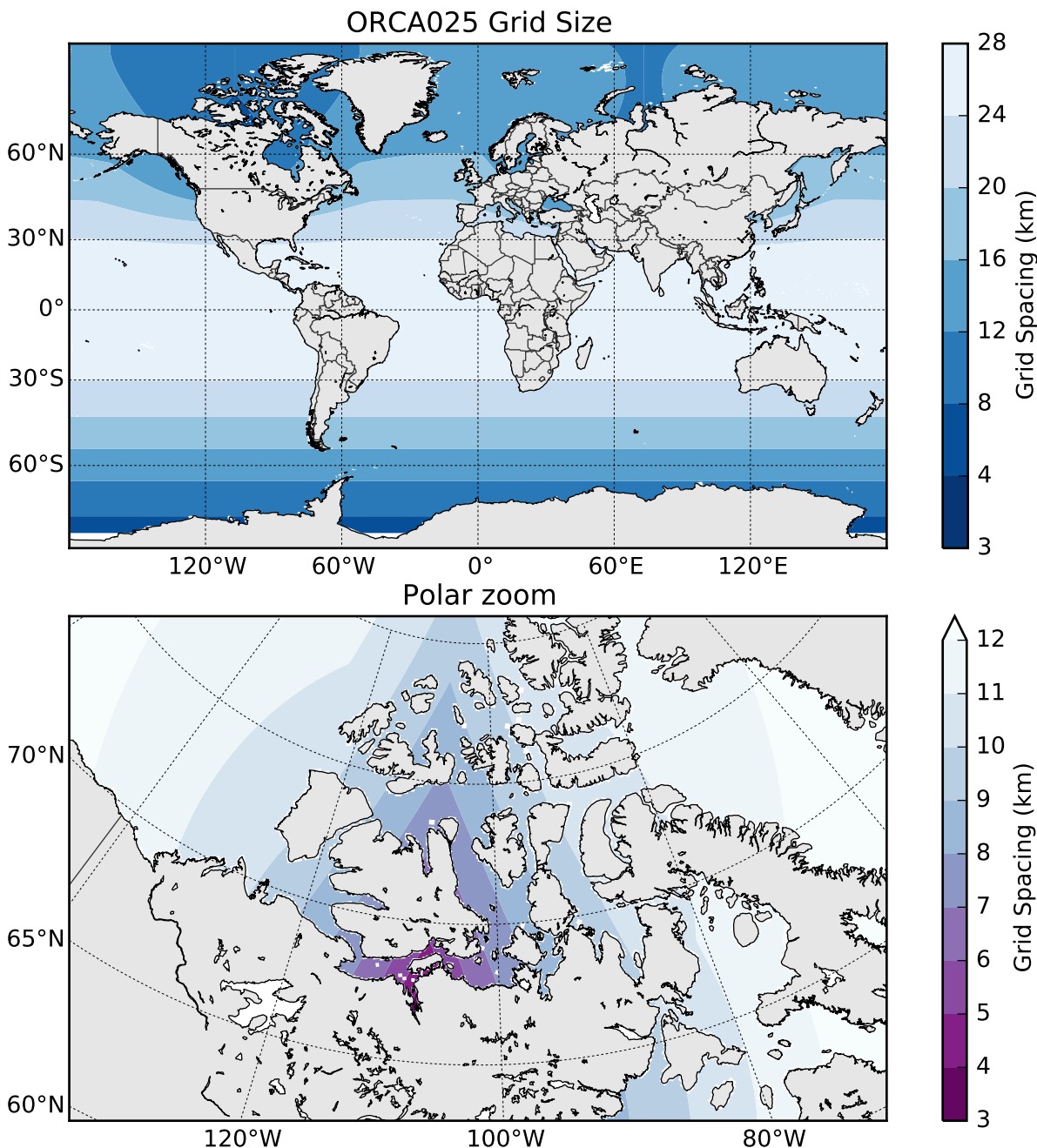

**Figure 1.** Grid size (defined as $\min(\texttt{e1t}, \texttt{e2t})$) on the ORCA025 grid. At top: in the global view, gridpoint spacing gradually decreases from the equator towards the north and south poles. At bottom: in a detail view of the north polar region, the grid is especially high-resolution in the southern portion of the Canadian Arctic Archipelago, with gridpoint spacing as low as 3km.

to regional models such as that of Lemieux et al. (2016), which refine this grid while retaining its tripolar structure to use conforming boundary conditions.

The coordinate system is also stretched in the vertical direction. Using the z-level grid option of the NEMO-OPA model, layers near the surface are spaced much more closely together than layers nearer the ocean bottom, in order to provide adequate resolution of the mixing layer. This stretching enhances the impact of vertical advection on the vertical Courant number, even if vertical-current magnitudes are low in absolute terms; Lemarié et al. (2015) notes that vertical advection provides a tighter bound on the timestep than horizontal advection.

Semi-Lagrangian advection alleviates both vertical and horizontal Courant number restrictions by tracing fluid parcels in a Lagrangian, fluid-following coordinate system. This coordinate system is defined so that the end of each timestep the fluid parcels arrive on the prescribed computational grid, and the properties of the fluid parcels (in the ocean setting, temperature, salinity, and horizontal velocity) at the end of the timestep (on the computational grid) are found by interpolating the previous-step, gridded values to the origin point of each parcel's trajectory. This method provides an implicit treatment of advection, allowing timesteps with advective Courant numbers greater than those usually permitted by explicit, Eulerian-form models.

In this work, we describe the initial implementation of a semi-Lagrangian advection routine in NEMO-OPA, based on the configuration of Smith et al. (2018). This configuration uses a linear free surface where the vertical coordinate does not move in time, but we believe that the described method can be generalized.

## 1.2  Existing work

In the atmosphere, semi-Lagrangian advection is a standard technique (Robert, 1982) for the implicit treatment of advection, but especially at large scales the effects of topography are relatively gentle. In particular, trajectory calculations can proceed under the assumption that the fluid parcel does not experience strong boundary-related acceleration. In the ocean domain this assumption is strongly violated, particularly for z-level vertical grids where the bathymetry changes abruptly at lateral cell boundaries.

Some attempts have been made previously to incorporate semi-Lagrangian advection into the ocean context. The work of Casulli and Cheng (1992), which is used as part of the ELCOM lake and estuary model (Hodges and Dallimore, 2006), calculates parcel trajectories via a substepping approach, where fluid parcel trajectories are integrated via an explicit Euler method over many short steps per model timestep. The two-dimensional, unstructured shallow water model of Walters et al. (2007) takes a similar approach, where it also must take at least one substep per element boundary traversed by a fluid parcel.

In this work, we overcome this difficulty with an iterative trajectory calculation which reduces in the limit to an implicit trapezoidal rule. In addition, we also derive the semi-Lagrangian advection scheme in a form which calculates effective advective tendencies, such that the advection routine can operate as a "plug-in" scheme for models which traditionally use Eulerian fluxes. We apply this to the NEMO-OPA model, and we believe this algorithm may be useful when applied to other ocean models with a structured grid.

In exchange, however, the semi-Lagrangian advection formulation departs from NEMO's finite-volume interpretation of its tracer and velocity components. By tracing infinitesimal fluid parcels, semi-Lagrangian advection treats gridpoint values

analogously to a finite-difference method, and as a consequence the scheme does not naturally offer conservation guarantees. This is not a primary concern for the short to medium-term forecasting applications that form the direct target for this work, but extensions of the semi-Lagrangian scheme to ensure conservation (Lauritzen, 2007) may be needed before the technique is applicable to longer-term climate simulations.

Additionally, Leclair and Madec (2011) has developed an "arbitrary Lagrangian-Eulerian" vertical coordinate scheme, implemented in recent versions of NEMO. This scheme splits vertical motions into fast (high temporal frequency) and slow motions, and the former are treated by co-moving vertical coordinate surfaces with a regridding step. This coordinate system reduces spurious diapycnal mixing caused by the high-frequency vertical motions, and its Lagrangian treatment of these motions relaxes the corresponding stability restriction.

## 1.3 Organization

We first introduce the time discretization of the semi-Lagrangian scheme in section 2, in order to develop a formulation that remains compatible with the common leapfrog scheme. In section 3, we begin to spatially discretize the semi-Lagrangian scheme by specifying the horizontal and vertical interpolation operators, and in section 4 we complete the discretization by defining the trajectory calculations. We present preliminary numerical examples in section 5, demonstrating the stability of the advection scheme.

## 2 Time discretization

The first requirement of a semi-Lagrangian advection scheme for the NEMO-OPA model is that it be consistent with the model's overall timestepping approach: the advection scheme is but one component of the full model.

In version 3 of NEMO-OPA, non-diffusive, non-damping processes such as advection are implemented via the leapfrog scheme (Mesinger and Arakawa, 1976), where at each timestep a field $f$ receives its new value at $f^A$ ($f$ "after" the timestep) based on its value at the previous timestep and forcing terms, which are all evaluated on the reference grid $\boldsymbol{x}_{ref}$. This gives a schematic of:

$$f^A(\boldsymbol{x}_{ref}) = f^B(\boldsymbol{x}_{ref}) + 2\Delta t \mathbf{F}(\boldsymbol{x}_{ref}), \tag{1}$$

where $f^A$ is the field calculated at time $t_0 + \Delta t$, $f^B$ is the field evaluated at the known prior time $t_0 - \Delta t$ ("before"), $f^N$ is the field at the provided time $t_0$ ("now"), and $\mathbf{F}$ is the forcing operator. The forcing operator includes advective processes at the "now" time-level, but diffusive, damping, and hydrostatic pressure terms might be evaluated at either the "before" or "after" time-levels.

This is an Eulerian approach to fluid motion, where tracer and momentum values are tracked along the fixed reference grid at all times, and fluid flows through this grid.

## 2.1 Semi-Lagrangian advection

In contrast, the Lagrangian advection schemes consider the fluid parcel to be the fundamental unit of discretization. In this perspective, if $f$ is a property of a fluid parcel that is conserved along a trajectory[1], it satisfies the continuous equations:

$$\frac{D}{Dt}f(\boldsymbol{x}(t)) = \mathbf{F}_L(\boldsymbol{x}(t)), \tag{2}$$

where $\frac{D}{Dt} = \partial_t + \boldsymbol{u} \cdot \nabla$ is the material derivative and $\mathbf{F}_L$ (Lagrangian right-hand side) contains all the same forcing terms as $\mathbf{F}$
*except* those arising from tracer and momentum flux, which are included inside the material derivative.

Ordinarily, (2) is discretized so that $\mathbf{F}_L$ is evaluated following the Lagrangian particles in the moving coordinate frame $\boldsymbol{x}(t)$, satisfying the trajectory equation:

$$\frac{D}{Dt}\boldsymbol{x}(t) = \boldsymbol{u}(\boldsymbol{x}(t)). \tag{3}$$

From an Eulerian point of view, (3) is a trivial identity based on the definition of the material derivative, but from the Lagrangian
point of view (3) must be solved to define $\boldsymbol{x}$ over time.

One technique for solving (2) and (3) is the two time-level implicit semi-Lagrangian method, used in the GEM atmospheric model (Girard et al., 2014) among others. Here, the $\mathbf{F}_L$ terms are evaluated with a trapezoidal rule, discretizing (2) and (3) as:

$$f^A(\boldsymbol{x}_{ref}) = f^N(\boldsymbol{x}^D) + \frac{\Delta t}{2}\big(\mathbf{F}_L^A(\boldsymbol{x}_{ref}) + \mathbf{F}_L^N(\boldsymbol{x}^D)\big) \text{ and} \tag{4a}$$

$$\boldsymbol{x}_{ref} = \boldsymbol{x}^D + \frac{\Delta t}{2}\big(\boldsymbol{u}^A(\boldsymbol{x}_{ref}) + \boldsymbol{u}^N(\boldsymbol{x}^D)\big). \tag{4b}$$

The trajectory equation (4b) acts to implicitly define the paths of the traced fluid parcels, where each location on $\boldsymbol{x}_{ref}$ is associated with a corresponding departure-point location $\boldsymbol{x}^D$. Over the single timestep, fluid parcels depart from $\boldsymbol{x}^D$ (which in general is not aligned with the grid) and arrive on the reference grid.

This off-grid, departure point evaluation of $\boldsymbol{u}$ and $\mathbf{F}_L$ is fundamental to Lagrangian and semi-Lagrangian methods, and $f^N(\boldsymbol{x}^D)$ ($\mathbf{F}_L^N(\boldsymbol{x}^D)$) can be written more simply as $f^D$ ($\mathbf{F}_L^D$) for "departure-point $f$ ($\mathbf{F}$)." Neither the time-implicit evaluations
(generally) nor the off-grid evaluations (of non-advective forcing) are compatible with the core structure of NEMO-OPA, which considers advection to be just one of many independent operators influencing the $\mathbf{F}$ term of (1).

## 2.2 Reconciliation

Implementing semi-Lagrangian advection in NEMO-OPA requires adopting as much of the framework of (1) as possible, without changing the evaluation of non-advective forcing terms. Effectively, the semi-Lagrangian advection routine must ultimately
supply a time-trend that, from the perspective of the leapfrog timestep algorithm, is indistinguishable from a conventional flux-form advection operator.

---

[1]This is true for temperature, salinity, and momentum provided the ocean is treated as an incompressible fluid. This assumption is satisfied by NEMO-OPA's adoption of the Boussinesq approximation.

To effect this, consider (2) without forcing terms ($\mathbf{F}_L = 0$). The function $f$ is preserved following the flow, so this gives the simply-written:

$$f^A = f^D. \tag{5}$$

This is approximated by taking one timestep of (1) (with only advective forcing $\mathbf{F}_{adv}$), but the latter involves integrating over the whole interval from $t_0 - \Delta t$ to $t_0 + \Delta t$. Thus, we should identify $f^D$ (and the departure points generally) not with the "now" time-level in the leapfrog scheme, but with the "before" time-level. Doing so and equating (5) and (1) gives:

$$f^A = f^B + 2\Delta t \mathbf{F}_{adv} = f^B(\boldsymbol{x}^D), \text{ or} \tag{6}$$

$$\mathbf{F}_{adv} = \frac{1}{2\Delta t}(f^B - f^B(\boldsymbol{x}^D)). \tag{7}$$

Equation (7) is prescriptive, and it gives the necessary trend for the leapfrog algorithm. Evaluating it requires $f$ only at the already-known "before" time-level and calculation of the departure points $\boldsymbol{x}^D$. This calculation is further simplified by basing the departure points on the time-centered velocities $\boldsymbol{u}^N$, and the exact algorithm for this calculation will be discussed in more detail in section 4.

## 2.3 Effects of the Asselin filter

To prevent decoupling of odd and even timesteps (damping the computational mode), NEMO-OPA is typically configured to use the Asselin time filter (Asselin, 1972), which adds a small time-damping proportional to $\frac{\partial^2}{\partial t^2} f$. Using the notation of Shchepetkin and McWilliams (2005) adapted to (1), the filter extends the time-marching scheme to the sequence:

$$f^{A*} \leftarrow f^B + 2\Delta t \mathbf{F}^{N*} \tag{8a}$$

$$f^N \leftarrow \epsilon f^{A*} + (1 - 2\epsilon)f^{N*} + \epsilon f^B \tag{8b}$$

$$f^{N*} \leftarrow f^{A*} \tag{8c}$$

$$f^B \leftarrow f^N \tag{8d}$$

Equation (8a) is the direct equivalent of (1), creating a provisional "after" value $f^{A*}$. Equation (8b) applies the filter (with a strength parameter $\epsilon$) with this value and the previous step's provisional field to define a final "now" field, and finally equations (8c) and (8d) are "bookkeeping" steps to shift field labels to become ready for the next timestep. The forcing operator $\mathbf{F}^{N*}$ is evaluated based on the provisionally-defined fields.

In applying this filter with the semi-Lagrangian forcing, equation (7) is oblivious to the presence of the filter or the difference between $f^N$ and $f^{N*}$. Substituting (7) into (8a) and applying (8c) and (8d) to (8a) and (8b) gives the update equation:

$$\begin{pmatrix} f^{N*} \\ f^B \end{pmatrix} \leftarrow \begin{pmatrix} f^B(\boldsymbol{x}^D) \\ \epsilon(f^B(\boldsymbol{x}^D) + f^B(\boldsymbol{x}_{ref})) + (1 - 2\epsilon)f^{N*}(\boldsymbol{x}_{ref}) \end{pmatrix}. \tag{9}$$

In the case of one-dimensional advection by a constant velocity $u_0$, the trajectory calculation is trivial and:

$$\boldsymbol{x}^D = x^D = x_{ref} - 2\Delta t u_0. \tag{10}$$

Since (9) is linear, we can also take its Fourier decomposition in space and consider only a single, arbitrary wave mode, giving $f = \hat{f}(t) \exp \mathbf{i}kx$ for a time-varying coefficient $\hat{f}$. Applying this to (9) casts the update in a matrix form as:

$$
\begin{pmatrix} \hat{f}^{N*} \\ \hat{f}^B \end{pmatrix} \leftarrow \begin{pmatrix} 0 & \exp(-2\mathbf{i}ku_0\Delta t) \\ 1 - 2\epsilon & \epsilon(1 + \exp(-2\mathbf{i}ku_0\Delta t)) \end{pmatrix} \begin{pmatrix} \hat{f}^{N*} \\ \hat{f}^B \end{pmatrix}. \tag{11}
$$

The time-stability of this filter is then governed by the eigenvalues of this matrix. Using the shorthand $\omega = -ku_0\Delta t$, these eigenvalues are:

$$
\lambda_{1,2} = \frac{1}{2}\left(\epsilon(1 + \exp(2\mathbf{i}\omega)) \pm \sqrt{\epsilon^2(1 + \exp(2\mathbf{i}\omega))^2 + (4 - 8\epsilon)\exp(2\mathbf{i}\omega)}\right), \tag{12}
$$

and to leading order in $\epsilon$ these eigenvalues have squared magnitudes of:

$$
|\lambda_{1,2}|^2 = 1 - 2\epsilon \pm 2\epsilon\cos(\omega) + O(\epsilon^2), \tag{13}
$$

signifying stability ($|\lambda| \leq 1$) for all values of $\omega$ and thus all Courant numbers.

## 3 Interpolation

To perform the off-grid interpolations in (7) to find $f^B(\mathbf{x}^D)$, this method fits a cubic polynomial to the underlying function. If the single departure point $\mathbf{x}_d = (x_d, y_d, z_d)$ lies within[2] $(x_a, x_{a+1}) \times (y_b, y_{b+1}) \times (z_c, z_{c+1})$ for integer values of $a$, $b$, and $c$ coinciding with grid-point locations, the full interpolation stencil consists of the grid-index cube $i \in [a-1, a+2]$, $j \in [b-1, b+2]$ and $k \in [c-1, c+2]$.

This grid-cube contains up to 64 grid points where $f(\mathbf{x})$ might be defined (subject to boundary conditions), and building a complete interpolation stencil would be cumbersome and inefficient. Instead, the interpolation procedure takes advantage of the tensor-product nature of the grid to separate interpolation along each dimension:

**Algorithm 1.** *Three-dimensional interpolation. To find $f(x_d, y_d, z_d)$ for some off-grid point $(x_d, y_d, z_d)$:*

1. *Interpolate $f(\mathbf{x})$ in the vertical to the location $[x_i, y_j, z_d]$, for $i \in [a-1, a+2]$ and $j \in [b-1, b+2]$*

2. *Interpolate along the first dimension in this two-dimensional grid to give $f(x_d, y_j, z_d)$, for $j \in [b-1, b+2]$.*

3. *Finally, interpolate along the second dimension to give $f(x_d, y_d, z_d)$.*

To effect the one-dimensional interpolations in algorithm 1, we make use of the cubic Hermite polynomials (Hildebrand, 1974). On the interval $0 \leq \chi \leq 1$, these polynomials are:

$$
\begin{aligned}
h_{00}(\chi) &= 2\chi^3 - 3\chi^2 + 1, \\
h_{01}(\chi) &= -2\chi^3 + 3\chi^2, \\
h_{10}(\chi) &= \chi^3 - 2\chi^2 + \chi, \text{ and} \\
h_{01}(\chi) &= \chi^3 - \chi^2,
\end{aligned} \tag{14}
$$

---

[2]If $\mathbf{x}_d$ lies along an edge or corner of this interval, then at least one of the resulting interpolations will be trivial. In that case, the choice of which neighbouring interval $\mathbf{x}_d$ lies "within" is arbitrary.

and a function $f(\chi)$ defined on this interval is interpolated via:

$$f(\chi) \approx f(0)h_{00}(\chi) + f'(0)h_{10}(\chi) + f(1)h_{01}(\chi) + f'(1)h_{11}(\chi). \tag{15}$$

Here, we prefer to use the cubic Hermite polynomials over simple Lagrange polynomial interpolation because the former choice allows greater freedom (via (15)) in implementation. If $f'$ is approximated by a four-point finite difference stencil, then (15) reduces to Lagrange interpolation. However, we can also make other choices for $f'$ to impose desirable properties: restricting $f'$ to have the same sign as the discrete difference imposes a type of slope limiting, and calculating $f'$ through a three-point stencil provides for continuous derivatives. These approaches are discussed in more detail in the following sections.

Interpolation using the above algorithm involves appropriately defining the interval to be scaled to $[0, 1]$ and approximating $f'$ at the endpoints. Because of the high aspect ratio of oceanic flows and the special character of vertical motion in a stratified ocean, these approximations differ between the horizontal and vertical interpolations.

## 3.1 Horizontal interpolation

In the horizontal, the interpolation in (15) can be directly conducted in grid-index space. Even when the underlying grid is mapped to the sphere, such as in the ORCA global grid (Madec, 2008, Ch. 16), the grid generally transitions smoothly and slowly from point to point[3]. The physical trajectory departure location $x_d$ (and $y_d$) can be translated into a fractional grid index offset by dividing by the appropriate grid scale factor, available inside the NEMO-OPA source code as one of `e[12][tuv]`.

Achieving third-order accuracy inside (15) is possible, but doing so requires an equally-accurate estimate for $f'$. Unfortunately, interpolating successively in one dimension using the above algorithm does not allow for precomputation of these derivatives: after the vertical interpolation step, all of the function values need to be taken off-grid, so any precomputed derivatives would themselves require interpolation. Instead, sufficiently-accurate estimates of the derivative are available by applying a finite-difference formula to the function values themselves.

## Derivative estimates

For notational simplicity, begin with the last step of the above algorithm where we have $f(x_d, y_j, z_d)$ and would like to estimate $f(x_d, y_d, z_d)$. If $y_d$ lies between $y_0$ and $y_1$, then the four-point interpolation stencil implies that we have computed $f(x_d, y_j, z_d)$ for $j = -1, 0, 1, 2$. To emphasize that this is now a one-dimensional interpolation problem, let $g(j) = f(x_d, y_j, z_d)$, such that $f(x_d, y_d, z_d) = g(j')$ for some $j' \in [0, 1]$. In this domain, $g'(0)$ and $g'(1)$ can be approximated by the finite differences:

$$g'(0) \approx -\frac{1}{3}g(-1) - \frac{1}{2}g(0) + \quad g(1) - \frac{1}{6}g(2) \text{ and} \tag{16a}$$

$$g'(1) \approx \frac{1}{6}g(-1) - \quad g(0) + \frac{1}{2}g(1) + \frac{1}{3}g(2), \tag{16b}$$

which then substitute for the appropriate derivatives in (15).

---

[3]This is not necessarily the case, however, for grids that have manually-specified, non-smooth regions of enhanced resolution. In such cases a more nuanced treatment of interpolation would be advisable.

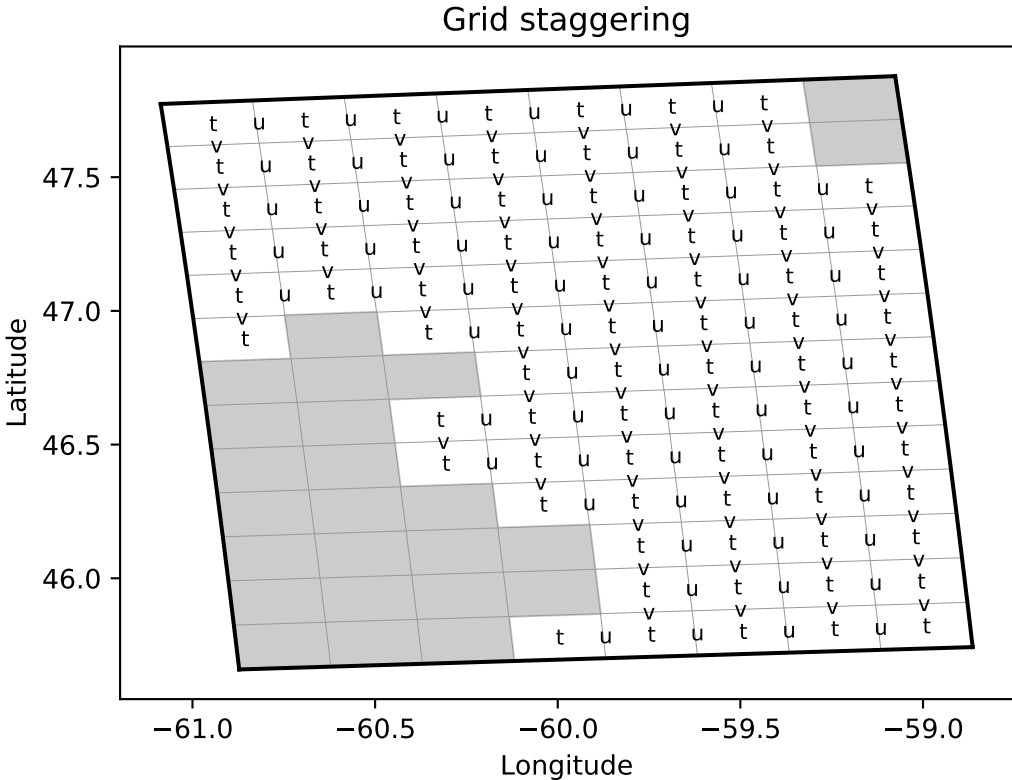

**Figure 2.** Grid point locations (letters) and land region (grey region) for a sample horizontal plane in the Gulf of Saint Lawrence, between Nova Scotia and New Brunswick. The horizontal velocities ($u$ and $v$) are staggered with respect to temperature and salinity ($t$), and the edge of the land area is coincident with the lines between velocity-point locations.

These finite differences are exact expressions for the first derivative for polynomials up to third order in $j$, and their use essentially converts (15) to interpolation via Lagrange polynomials. The Hermite polynomial form, however, allows for an
easier imposition of boundary conditions.

**Boundary conditions**

On the NEMO-OPA $z$-level grid, the lateral boundaries coincide with $u$- and $v$-points (velocity points), which are spaced halfway between $t$-points (tracer points). Tracer points that lie inside the land region are masked ($\texttt{tmask} = 0$) as are velocity points that are at the edge of or within the masked region. This arrangement is illustrated for a sample region in figure 2.
The physical interpretation of the boundary varies with respect to the field being interpolated. For tracers, lateral boundaries imply no-flux conditions for the purposes of advection, which in turn implies a zero derivative at the boundary. The normal velocity ($u$ with respect to a boundary along the first grid dimension, $v$ with respect to a grid boundary along the second) is

obviously constrained to zero by geometry to give a Dirichlet boundary condition, whereas the tangential velocity can be set as free-slip, no-slip, or some combination via a namelist entry. In the subsequent, we assume that velocity has a free-slip boundary condition, with boundary friction left for future work.

If a boundary occurs in the left portion of this interpolation stencil, there are a total of seven possible cases:

**Algorithm 2.** *Lateral boundary conditions*

*To find $g(j')$ for $0 \leq j' \leq 1$ in the vicinity of a lateral boundary:*

1. *If $g(-1)$ corresponds to a point at the boundary and the boundary is of the Dirichlet-type, then $g(-1) = 0$ and (16a) and (15) apply normally.*

2. *If $g(-1)$ is inside the boundary, $g(0)$ is inside the fluid domain (that is, the boundary is between these two points), and the boundary is of the Neumann-type, then $g(-1)$ is taken to be equal to $g(0)$, essentially making it a ghost point.*

3. *If $g(-1)$ is inside the boundary, $g(0)$ is inside the fluid domain, and the boundary is of the Dirichlet-type, then $g(-1)$ is taken to be $-g(0)$.*

4. *If $g(0)$ is at the boundary and the boundary is of the Dirichlet-type, then $g(0) = 0$ and $g(-1) = -g(1)$.*

5. *If $g(0)$ is inside the boundary and $j' < 0.5$, then the interpolated point is itself inside the boundary and should be masked.*

6. *If $g(0)$ is inside the boundary, $j' > 0.5$, and the boundary is of the Neumann-type, then $g(0) = g(1)$ and $g(-1) = g(2)$.*

7. *If $g(0)$ is inside the boundary, $j' > 0.5$, and the boundary is of the Dirichlet-type, then $g(0) = -g(1)$ and $g(-1) = -g(2)$.*

For boundaries that occur in the right portion of the interpolation stencil, the values taken for ghost points are given symmetrically.

The combination of "the grid point is at the boundary" and "the boundary is of Neumann-type" is missing from algorithm 2. This construction is forbidden by the grid structure of NEMO-OPA, where tangential velocity is located one half-cell away from a boundary.

For two dimensional interpolation, algorithm 2 applies independently to each dimension. When interpolating along $x$, the points $f(x_d, y_j, z_d)$ will each individually be either in the ocean domain and valid or in the land domain and masked, which provides the values necessary to compute $f(x_d, y_d, z_d)$. This off-grid point itself must lie in water, which imposes a strong requirement on the trajectory calculations to be discussed in section 4.

**Slope limiting**

As a final step, once values for the function and its derivative at the interval endpoints are specified, the derivative values are limited to help prevent new maxima in the interpolated function. In particular, if $g(0)$ is a local minimum (maximum) among

itself, $g(-1)$, and $g(1)$, then $g'(0)$ is set to zero if the above procedure finds that it would be negative (positive). A similar procedure applies symmetrically for $g'(1)$ if $g(1)$ is a relative extremum.

This limiting is milder than methods derived from Bermejo and Staniforth (1992), which would strictly preserve positivity for any $j'$, but it effectively limits excursions when $j'$ is close to $0$ or $1$. Without such limiting, numerical testing showed that semi-Lagrangian advection of temperature and salinity could cause weak instabilities near the coastline, where a locally extreme temperature or salinity could become "trapped" near the coast and slowly amplified.

### 3.2 Vertical interpolation

Vertical motion in the NEMO-OPA model differs from horizontal motion in a number of respects:

- Vertical gradients of temperature and density are much stronger than typical horizontal gradients, especially near the surface.

- Typical vertical grids used with NEMO-OPA are strongly stretched, with a higher resolution near the surface and a lower resolution in the deep ocean.

- Vertical flow is often oscillatory, where vertical motion is driven by barotropic and baroclinic waves.

The horizontal interpolation described in section 3.1 is third-order accurate; with the provided one-sided formulas for calculating the endpoint derivatives it reduces to a four-point (cubic) Lagrangian interpolation process. However, the smooth field implied by this interpolation process is only $C^0$ continuous: $f(x_j - \epsilon)$ "sees" $f'_j$ calculated from $f(x_{j-2})$ to $f(x_{j+1})$, whereas $f(x_j + \epsilon)$ sees $f'_j$ from $f(x_{j-1})$ to $f(x_{j+2})$.

We do not find this to be a practical concern for horizontal interpolation, since horizontal currents in most of the ocean tend to be dominated by relatively steady quasi-geostrophic motions. In the vertical, however, we found that even low-amplitude oscillations caused by high-frequency gravity waves would cause the temperature and salinity fields to drift. The mechanism is that a fluid parcel displaced upwards by $\epsilon$ in one timestep and downwards by $\epsilon$ in the next timestep would see an effective diffusion proportional to the jump between the upward and downward-looking vertical derivatives.

To maintain global accuracy, we impose $C^1$ continuity in the vertical direction through an alternative treatment of the vertical derivative. Instead of applying equations (16), we treat the physical depth (rather than grid index) as the relevant coordinate and construct a centered estimate of the derivative.

For a function $f(z_n)$ defined at the $z_n$ levels, define $\Delta f^+ = f(z_{n+1}) - f(z_n)$, $\Delta f^- = f(z_n) - f(z_{n-1})$, $\Delta z^+ = z_{n+1} - z_n$, and $\Delta z^- = z_n - z_{n-1}$. These differences combine to give the estimated derivative:

$$f_z(z_n) \approx \frac{1}{\Delta z^- + \Delta z^+} \left( \frac{\Delta z^-}{\Delta z^+} \Delta f^+ + \frac{\Delta z^+}{\Delta z^-} \Delta f^- \right), \tag{17}$$

which is accurate to $O(\Delta z^2)$ for the derivative and accurately reproduces quadratic functions of $z$. In the limiting case of a constant $\Delta z$ (equispaced vertical levels), this formula reduces to the classic centered difference.

Because vertical interpolation comes first in algorithm 1, (17) need be evaluated only at grid points, and in fact it may be precomputed for the entire grid for a given function and timestep. This is a key advantage of placing vertical interpolation first in the interpolation sequence, and it avoids duplication of work.

Whereas interpolation near the horizontal boundaries is complicated by the many combinations of grid staggering and physical boundary conditions, interpolation near the vertical boundaries is much simpler. On the NEMO grid, tracers and horizontal velocities lie along the same vertical level, and these levels are staggered one half-cell away from the boundaries. Likewise, the natural vertical boundary condition for both tracer and horizontal velocity fields is a no-flux boundary condition; NEMO-OPA models boundary-layer friction in another module. Interpolation near the boundaries then proceeds in two steps.

The first step is to define $f_z$ at the top and bottom points in the water column, for which the central difference formula of (17) is not directly valid. Here, we approximate the physical no-flux condition through a fictitious ghost point such that $\Delta f^- = 0$ at the top boundary and $\Delta f^+ = 0$ at the bottom boundary, with the respective $\Delta z$ matching the layer thickness (e3t).

The second step is to define how (15) applies to the interval between the grid level and the physical boundary. Here, the no-flux boundary conditions reduce to even symmetry, and the derivative at the ghost points is the negative of the vertical

derivative calculated for the in-boundary point. Near the free surface, if the interpolation point is above the level of the free surface (above $z = 0$) then it is clamped to the surface itself. Near the ocean bottom, if the interpolation point is below the level of the ocean bottom (below $z = z_{max}$) then the point is masked and is treated as an "inside the boundary" point for the purposes of horizontal interpolation above.

**Treatment of partial cells**

Over most of the domain, this interpolation works well. Although there is no guarantee of positivity in the derivative formulation of (17), overshoots and the consequent generation of spurious maxima are limited. For the tests presented in section 5, there was no need for slope-limiting for vertical interpolation over most of the domain.

One exception to this rule is at the bottom boundary. Here, vertical levels are spaced far apart, but to better-represent the ocean bottom the z-level grid of NEMO-OPA uses a partial cell configuration (Madec, 2008, sec. 5.9). For water columns where

the bottom-most cell is much deeper than its neighbours, a local (small) upwelling can cause an overshoot of temperature or salinity that spuriously increases the local density but does not diminish the upwelling. Over time, the maxima-increasing trend can accumulate and cause some points at the bottom boundary to reach implausibly cold temperatures (below $-10°\,\text{C}$, for example) or high salinities. In the absence of explicit horizontal diffusion (which would mix this maximum into more dynamically-active regions), these spurious maxima do not generally corrupt the flow, although they obviously would corrupt

whole-ocean (or whole-level) statistics such as average or extreme temperatures.

Near these boundary cells, vertical limiting is implemented in the simplest possible way: the interpolation of (15) is replaced with a constant, such that $f(z) = f(z_k)$ over the interval from $z_k$ downwards to the physical boundary.

Implementing this limiting over the whole bottom level is possible, but that is far stronger than necessary and leads to erroneous diffusion along gentle slopes. When the bottom layer is composed of partial cells of varying thickness, even inter-

polation along a horizontal plane (that is, without changing physical depth) requires vertical interpolation in grid space to find

that constant level in adjoining columns. Imposing vertical limiting along the whole bottom level effects undesired horizontal diffusion, even though the problem solved by limiting is observed when adjoining cells have large relative thickness variations.

As a compromise between these two errors, we only apply the described limiting to vertical interpolation for cells at the bottom boundary which have a layer thickness greater than $1.75$ times that of their "thinnest" neighbour.

This exact threshold is empirical, and other grids might require a re-tuning of this parameter. Ideally, the grid generation itself would avoid abrupt transitions in cell-layer thicknesses, but adding such a restriction would make this advection scheme useless as a drop-in replacement for the standard advection routines of NEMO-OPA.

### 3.3 A numerical example

As a simple numerical example, consider the case of a tracer being advected in a rectangular, two-dimensional domain by an
340 internal wave and a background current. This tracer satisfies the advection equation:

$$\frac{\partial \sigma}{\partial t} - u(x,z,t)\frac{\partial \sigma}{\partial x} - w(x,z,t)\frac{\partial \sigma}{\partial z} = 0, \tag{18}$$

for some prescribed velocity field $(u, w)$.

If this tracer field $\sigma(x,z,t)$ would be a function of $z$ alone ($\bar{\sigma}(z)$) if not for the wave motion, then its motion is analytically given by:

$$\sigma(x,z,t) = \bar{\sigma}\big(z - \eta(x - (c + u_0)t, z)\big), \tag{19}$$

where $\eta(x,z)$ is the isopycnal displacement, $u_0$ is the $x$-directed background current (uniform in $z$), and $c$ is the phase speed of the wave. Following Turkington et al. (1991), a streamfunction defined as:

$$\psi(x,z,t) = c\eta(x,z,t) - u_0 z \tag{20}$$

gives velocities:

$$u = -\psi_z \tag{21a}$$

$$w = \psi_x, \tag{21b}$$

which are exact solutions of (18) for $\sigma(x,z,t)$.

To give an internal wave that respects no-flux conditions at the top and bottom of the domain, we set:

$$\eta(x,z,t) = A\cos\big(k(x - (c + u_0)t)\big)\sin(mz) \tag{22}$$

where $k$ and $m$ are horizontal and vertical wavenumbers respectively and $A$ is the wave amplitude. For a domain of size $L_x$ in the horizontal (periodic) and $L_z$ in the vertical, $k = 2\pi/L_x$ and $m = \pi/L_z$ give the lowest internal wave mode, used here.

In dimensional units, we take the model domain to be a channel $L_x = 1$km long and $L_z = 100$m deep with a background current of $u_0 = 1\mathrm{m\,s}^{-1}$, and we set $c = N/\sqrt{k^2 + m^2}$ based on a mean buoyancy frequency of $N = .03\mathrm{s}^{-1}$, which corresponds

to a 1% density change from the surface to the bottom of the channel. With a wave amplitude of $A = 10\text{m}$, the maximum wave-induced current is about 10% of $u_0$, and the phase speed is $c \approx 0.94\text{ms}^{-1}$.

In order to represent the pycnocline found in many ocean waters, we choose[4] $\bar{\sigma}(z) = \tanh\left(\frac{1}{2} - \frac{1}{10}zL_y^{-1}\right)$.

The domain is discretized by $N_x \times N_y$ points, defined as:

$$x_i = -\frac{L_x}{2} + L_x \frac{i - 0.5}{N_x}, \text{ and} \tag{23a}$$

$$z_j = \frac{L_z}{2}\left(1 + \frac{\alpha_j + \alpha_j^3}{2}\right), \tag{23b}$$

where $i = 1, 2, \cdots, N_x$, $j = 1, 2, \cdots, N_z$, and:

$$\alpha_j = 2\frac{j - 0.5}{N_z} - 1. \tag{23c}$$

This implements a stretched vertical coordinate that increases the vertical resolution in the vicinity of the pycnocline.

**Semi-Lagrangian advection**

In integrating this system with semi-Lagrangian advection, the leapfrog method reduces to an Euler method of twice the timestep because there is no external forcing. The time-discrete equation is:

$$\sigma(x_i, z_j, t + 2\Delta t) = \sigma(x_{d(ij)}, z_{d(ij)}, t), \tag{24}$$

where $(x_{d(ij)}, z_{d(ij)})$ is the departure point of the trajectory that arrives at the gridpoint $(x_i, z_j)$, and the off-grid evaluation of $\sigma$ proceeds via the interpolation processes described earlier without slope-limiting.

The departure points are given by the trapezoidal rule[5] with a time-centered evaluation of velocity:

$$x_i - x_{d(ij)} = \Delta t\left(u\left(x_i, z_j, t + \Delta t\right) + u\left(x_{d(ij)}, z_{d(ij)}, t + \Delta t\right)\right) \tag{25a}$$

$$z_j - z_{d(ij)} = \Delta t\left(w\left(x_i, z_j, t + \Delta t\right) + w\left(x_{d(ij)}, z_{d(ij)}, t + \Delta t\right)\right), \tag{25b}$$

where the velocities are evaluated exactly via (21). The overall system (25) is solved via simple iteration, with an initial guess given by setting $(x, z)_{d(ij)} = (x, z)_{ij}$.

This algorithm is stable for large timesteps, so we tested this system for timesteps corresponding to Courant numbers of 0.2 and 2.1, with spatial grid resolutions between $40 \times 4$ and $2560 \times 256$. The final integration time was chosen to be $t_{fin} = 5L_x/u_0$, which allowed the wave to propagate through the domain several times. Since the exact solution is analytically known, we recorded the maximum error experienced over the integration, and error convergence rates are shown in figure 3.

---

[4]Since this section tests advection alone, the scaling of $\sigma$ is not dynamically relevant. In fact, the wave structure of (22) corresponds to an exact internal mode of the incompressible Navier-Stokes equations for a linear stratification.

[5]The trajectory calculation scheme of section 4 could be used instead, but since the overall trajectory lengths are small compared to the length scales of the velocity field ($L_x$ and $L_z$), that method would give equivalent results to the trapezoidal rule.

**Flux-form advection**

As a control, we also integrate this system in flux form ($\sigma_t - \nabla \cdot (\boldsymbol{u}\sigma) = 0$) via centered differences, with $\sigma$ evaluated at the midpoints between grid cells via a simple average, matching the central difference tracer advection scheme in NEMO-OPA.

The velocity field given by (20) is divergence free, so this form of the equation is pointwise equivalent to (18). However, this no longer holds after discretization. In order to eliminate the divergence error, the velocity field is defined by creating the streamfunction at the staggered points $(x_{i+1/2}, z_{j+1/2})$ and defining discrete velocities $u$ and $w$ via the discrete equivalents to (21). With this modification, the discrete flux-form operator is equivalent to a discrete advection equation.

After leapfrog discretization in time, the discretized equation is:

$$
\begin{aligned}
\sigma(x_i, z_j, t+\Delta t) = \sigma(x_i, z_j, t-\Delta t) + 2\frac{\Delta t}{\Delta x \Delta z_j} \Big( & \frac{\Delta z_j}{2}\big(u(x_{i-1/2}, z_j, t)(\sigma(x_{i-1}, z_j, t) + \sigma(x_i, z_j, t)) - \\
& u(x_{i+1/2}, z_j, t)(\sigma(x_i, z_j, t) + \sigma(x_{i+1}, z_j, t))\big) + \\
& \frac{\Delta x}{2}\big(w(x_i, z_{j-1/2}, t)(\sigma(x_i, z_{j-1}, t) + \sigma(x_i, z_j, t)) - \\
& w(x_i, z_{j+1/2}, t)(\sigma(x_i, z_j, t) + \sigma(x_i, z_{j+1}, t))\big)\Big),
\end{aligned}
\tag{26}
$$

where $\Delta z_j = z_{j+1/2} - z_{j-1/2} = \frac{1}{2}(z_{j+1} - z_{j-1})$. For the first timestep, a single Euler step is taken of size $\Delta t$ with time-centered velocities ($t = \Delta t/2$).

As usual, this leapfrog timestepping algorithm is only stable to a maximum Courant number of $1$. With this staggered grid and vertical grid stretching, the Courant number can be defined by:

$$
C_{ij} = \Delta t \left( \frac{\max(u_{i+1/2,j}, 0) - \min(u_{i-1/2,j}, 0)}{\Delta x} + \frac{\max(w_{i,j+1/2}, 0) - \min(w_{i,j-1/2}, 0)}{\Delta z_j} \right).
\tag{27}
$$

For the mode-one internal wave with background current used in this section, the maximum Courant number is reached at the top and bottom of the domain (where $w = 0$), so $\max(C) = \max(u)/\Delta x$.

We present results for (26) at a maximum Courant number of $0.2$, chosen to give a "small timestep" for later comparison with semi-Lagrangian results. The results are insensitive to the timestep within the stable range, with less than 5% change in maximum-norm error over the range $0.2 \leq \max(C) \leq 0.99$.

**Results**

The error over time of this test case is shown in figure 3. As expected, each method achieves second-order convergence. For the Eulerian advection control case, this is governed by its two-point central difference scheme. For the semi-Lagrangian cases, the dominant contribution to the error field comes from the lower-order vertical interpolation. While the semi-Lagrangian method has a higher order of accuracy for horizontal motion, here the problem is constructed such that horizontal and vertical motions are of equal importance.

As is often observed with semi-Lagrangian methods, the overall error of the scheme is somewhat lower for the high-CFL case than for the low-CFL case. The interpolation used to evaluate $\sigma$ off the grid introduces error with each interpolation, and

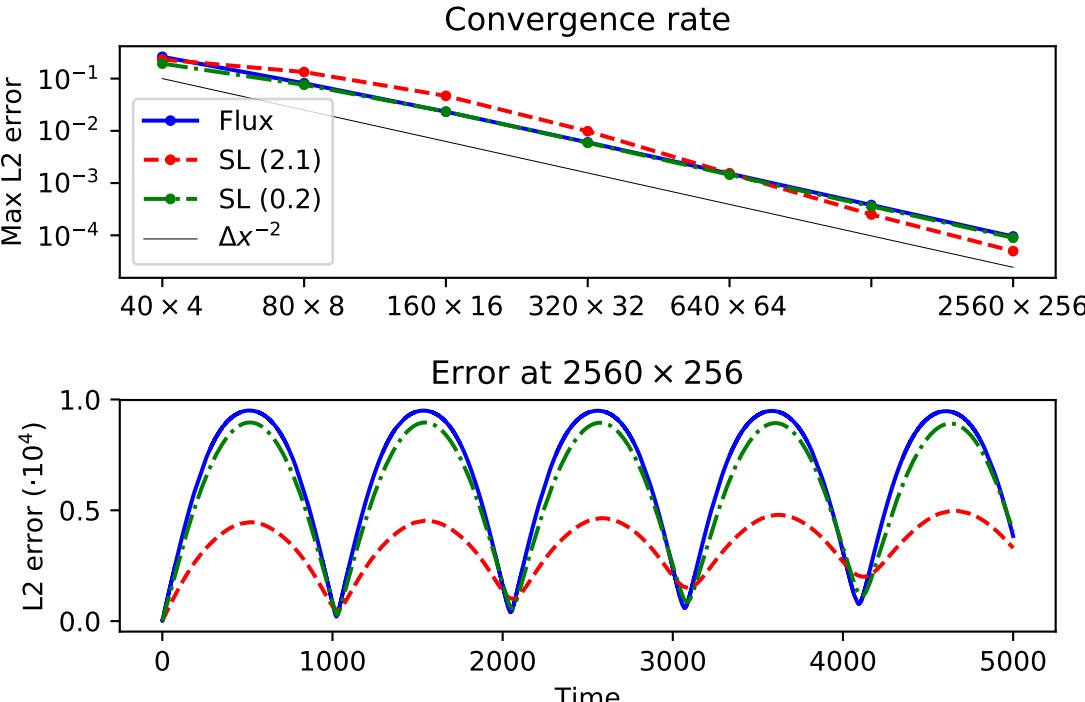

**Figure 3.** Top: Maximum L2 error $((\int (\sigma - \sigma_{ex})^2 dA/(L_x L_z))^{1/2})$ over the integration period for the test case of section 3.3 versus resolution for flux-form Eulerian advection with a Courant number of 0.2 (blue, solid), semi-Lagrangian advection with a Courant number of 2.1 (red, dashed), and semi-Lagrangian advection with a Courant number of 0.2 (green, dot-dashed), showing second-order convergence (line). Bottom: L2 error over time for these algorithms, on the $2560 \times 256$ grid.

the overall contribution of this error necessarily scales in proportion to the number of interpolations and inversely with the timestep size.

Overall, this simplified test case supports the conclusion that the semi-Lagrangian treatment of advection is a viable replacement for flux-form advection. The semi-Lagrangian method achieves similar (for low-CFL flows) or better (for high-CFL flows) error, and it remains stable for CFL values substantially larger than unity.

## 4   Trajectory calculation

With the mechanism for evaluating the $f^B(x^D)$ term in (7) established in section 3, the remaining half of the semi-Lagrangian advection algorithm is the estimation of the $x^D$ departure points. This corresponds to the positions at the "before" time-level $(t_0 - \Delta t)$ of those fluid parcels that will arrive on the reference grid $x_{ref}$ the "after" time-level $(t_0 + \Delta t)$. One such upstream

location exists for each valid grid location, so in general $\boldsymbol{x}_d$ needs to be estimated for each t, u, and v point on the NEMO-OPA grid to provide for (respectively) the tracer and velocity advective forcings.

In general, calculation of the departure points is an implicit and nonlinear problem, requiring knowledge of the flow velocity at every sub-grid place and time between the "before" and "after" time-levels, before the flow at the latter has been computed. To make this problem tractable, we make a series of simplifying assumptions.

The first such assumption is to freeze the flow, such that trajectories are computed based on strictly the "now" velocities (that is, $u$, $v$, and $w$ at the intermediate time-level). This is consistent with the underlying leapfrog timestepping algorithm and the other advection schemes in NEMO-OPA, where most fluxes are computed instantaneously with respect to the same "now" velocities. In physical terms, this constrains fluid parcels to follow paths based on estimated, instantaneous streamlines. In exchange, this decouples the trajectory computation from the "after" velocities and makes the process time-explicit, which eliminates what would otherwise be a need to iterate the entire timestepping process.

## 4.1 Exponential integration

Ordinarily, the next assumption in the trajectory calculation is to approximate the particle paths, either by a straight line or by a low-degree polynomial. In this case, the Lagrangian equation:

$$\frac{\mathrm{d}\boldsymbol{x}}{\mathrm{d}t} = \boldsymbol{v}(\boldsymbol{x}) \tag{28}$$

is integrated with an approximate quadrature. Using the trapezoidal rule gives the approximation:

$$\boldsymbol{x}_a - \boldsymbol{x}_d = \Delta t (\boldsymbol{v}(\boldsymbol{x}_d) + \boldsymbol{v}(\boldsymbol{x}_a)), \tag{29}$$

where $\boldsymbol{x}_a = \boldsymbol{x}(t_0 + \Delta t)$ is the on-grid "arrival" point of the fluid parcel. This approximation is second-order in time, and it results in an iterative method where $\boldsymbol{v}(\boldsymbol{x}_d)$ is interpolated, leading to a revised estimate of $\boldsymbol{x}_d$.

Unfortunately, this approximation is not suitable for trajectory calculations in the general ocean because it does not appropriately handle flow near a solid boundary. Consider the case of two-dimensional flow in the positive half-plane, where fluid velocities are prescribed as $(u, v) = (x, -y)$. This forms an analytic continuation of flow near a boundary along the line $x = 0$.

Now, apply equation (29) to the fluid parcel that arrives at $\boldsymbol{x}_a = (1, 0)$. Along this streamline, $v = 0$ by inspection so this equation reduces to one dimension and has the solution:

$$x_d = \frac{1 - \Delta t}{1 + \Delta t}. \tag{30}$$

For small values of $\Delta t$, this solution is reasonable. For $\Delta t > 1$, however, this solution leads an unphysical trajectory, where the departure point is found to lie in the left-half plane (and thus lies inside the boundary).

The failure here is a specific example of (29) failing the Lipschitz trajectory-crossing criterion (Smolarkiewicz and Pudykiewicz, 1992), which requires $u_x \Delta t < C \approx 1$. The trajectory implied by (30) crosses the trajectories of fluid parcels that arrive at $\boldsymbol{x}_a = (1 \pm \epsilon, 0)$, and the resulting advection loses its physical meaning.

This trajectory-crossing criterion is a physical limit for solutions which develop discontinuous shocks, such as those that can arise in simulations of the non-dispersive, nonlinear shallow water equations. However, these shocks are not typical of three-dimensional hydrostatic flows in the ocean, and they are certainly not universally seen at solid boundaries. The *true* trajectories of fluid parcels, if evaluated exactly, do not cross (and do not have origins inside the land domain), so a better approach is to directly integrate (28) without approximating the time derivative. Here, this one-dimensional system reduces to the ordinary differential equation:

$$x_t = x \tag{31}$$

with the boundary condition $x(t_0 + \Delta t) = x_a = 1$. The solution to this equation is obviously of the form $x(t) = C \exp(t)$ for some constant $C$, and applying the boundary condition gives $x(t) = \exp(t - (t_0 + \Delta t))$ and a departure point of $x_d = \exp(-2\Delta t)$.

This solution is very well-behaved, lying exclusively in the right half-plane and asymptotically approaching the wall at $x = 0$ as $\Delta t \to \infty$. This approach works when that of (29) fails because the direct integration properly captures the exponential-in-time path of the fluid parcel.

A generalization of this approach forms the basis for trajectory calculation in this work. Since the solution of (28) is not analytically possible with an arbitrary velocity field, we exactly solve (28) based on an approximate, linearly-varying velocity field. This is similar to an approach discussed by Walters et al. (2007), where within a single, two-dimensional finite-element cell the linear velocity form is exactly-given by the underlying discretization rather than an approximation.

Assume that an arbitrary fluid parcel arrives at $\boldsymbol{x}_a$, and that we know the velocity there ($\boldsymbol{v}_a$) and at another point $\boldsymbol{v}(\boldsymbol{x}_c) = \boldsymbol{v}_c$. We know that the fluid parcel must arrive at $\boldsymbol{x}_a$ travelling in the direction of $\hat{v}_a = \boldsymbol{v}_a / \alpha_a$, with $\alpha_a = \|\boldsymbol{v}_a\|$. Then $\boldsymbol{v}_c$ can be written in terms of this direction as $\boldsymbol{v}_c = \alpha_c \hat{v}_a + \beta_c \hat{n}_a$, for scalar $\alpha_c$ and $\beta_c$ and some $\hat{n}_a$ normal to $\hat{v}_a$.

This forms a two-dimensional system spanned by vectors $\hat{v}_a$ and $\hat{n}_a$. If we additionally make the assumption that $\boldsymbol{v}(\boldsymbol{x})$ varies linearly in this plane, we can construct a simplified, two-dimensional coordinate system to solve (28). Here, the origin of the coordinate system corresponds to $\boldsymbol{x}_a$, and the rotated coordinates $\hat{x}$ and $\hat{y}$ align with $\hat{v}_a$ and $\hat{n}_a$ respectively. This implies that $\boldsymbol{x}_c$ projects onto the point $(\boldsymbol{x}_c \cdot \hat{v}_a, \boldsymbol{x}_c \cdot \hat{n}_a) = (x_c, y_c)$. The linearly-interpolated velocities lie strictly in this plane, so the equations of motion for a fluid parcel are:

$$x_t = \alpha_a + (\alpha_c - \alpha_a)\frac{x}{x_c}, \text{ and} \tag{32a}$$

$$y_t = \beta_c \frac{x}{x_c}, \tag{32b}$$

subject to the boundary condition that $x(t_0 + \Delta t) = y(t_0 + \Delta t) = 0$. (32a) can be solved first, and applying the boundary condition $x(t_0 + \Delta t) = 0$ gives:

$$x(t) = \frac{\alpha_a x_c}{\alpha_c - \alpha_a}\left( \exp\left( \frac{\alpha_c - \alpha_a}{x_c}(t - (t_0 + \Delta t)) \right) - 1 \right). \tag{33a}$$

Applying this to (32b) along with its boundary condition $y(t_0 + \Delta t) = 0$ gives:

$$y(t) = \frac{\beta_c \alpha_a}{\alpha_c - \alpha_a}\left( \frac{x_c}{\alpha_c - \alpha_a}\left( \exp\left( \frac{\alpha_c - \alpha_a}{x_c}(t - (t_0 + \Delta t)) \right) - 1 \right) - (t - (t_0 + \Delta t)) \right). \tag{33b}$$

When the along-trajectory acceleration is small ($|(\alpha_c - \alpha_a)\Delta t / x_c| \ll 1$), (33) reduces to a trapezoidal rule with second-order accuracy in time.

**Trajectory iteration**

Evaluating (33) at $t = t_0 - \Delta t$ and re-projecting the coordinates to the grid forms the basis of an iterative algorithm for trajectories:

**Algorithm 3.** *Trajectory iteration overview*

 *At each grid point:*

  1. *Begin with a candidate departure point* $\boldsymbol{x}_c = \boldsymbol{x}_a - 2\Delta t \boldsymbol{v}_a$

  2. *Interpolate the "now" velocities off-grid to this point*

  3. *Evaluate* (33) *at* $t = t_0 - \Delta t$ *to give a revised candidate departure point* $\boldsymbol{x}'_c$

  4. *Set* $\boldsymbol{x}_c \leftarrow \boldsymbol{x}'_c$

  5. *Repeat from step 2 until the change is smaller than a tolerance of* $10^{-3}$ *grid cells*

This algorithm is ideally suited to cases that look like flow away from a stagnation point, where a fluid parcel is accelerating as it reaches the grid point at $t_0 + \Delta t$. In those cases, the $(\alpha_c - \alpha_a)/x_c$ terms will be positive, and the exponential terms will limit the size of the trajectory for finite $\Delta t$. In the opposite case, however, the exponential terms will tend to lengthen the trajectory. For large $\Delta t$ or large deceleration, this effectively demands that (32)–(33) extrapolate beyond the velocity sample at $\boldsymbol{x}_c$, a potential source of instability.

To remedy this, a limiter is added to step 3 of algorithm 3, whereby $x(t_0 - 2\Delta t)$ is constrained to the greater[6] of that from (33a) and $-2\Delta t \max(\alpha_a, \alpha_c)$. When limiting is necessary it effectively reduces the timestep used for the trajectory iteration, so for consistency a revised $\Delta t'$ is computed by inverting (33a) with the limited $x'_c$, which is then used to evaluate (33b).

## 4.2 Underrelaxation and land boundaries

While the construction of algorithm 3 guarantees that trajectories cannot converge to an out-of-boundary point, there are no guarantees that the algorithm remains in-boundary during the iteration process or that the iteration converges. The problem of a divergent or oscillatory iteration is more likely when the underlying velocity field does not resemble the linearly-approximated velocity field integrated by (32), as then each iteration might result in very different approximations.

Addressing the latter point first, this work heuristically applies underrelaxation when algorithm 3 is slow to converge. After 10 local iterations, step 3 is replaced by $\boldsymbol{x}_c \leftarrow \frac{1}{2}(\boldsymbol{x}_c + \boldsymbol{x}'_c)$, after 20 iterations the right-hand side becomes $\frac{1}{4}(3\boldsymbol{x}_c + \boldsymbol{x}'_c)$, and after 30 iterations the right-hand side becomes $\frac{1}{8}(7\boldsymbol{x}_c + \boldsymbol{x}'_c)$. At 40 iterations, the trajectory is truncated by ending the iteration

---

[6]Since the rotated x-axis is aligned with the fluid velocity at $\boldsymbol{x}_a$, $x_c$ is generally negative in the rotated frame.

with the first in-domain point returned from the process; this ensures some sort of advection even if the iterative process enters a limit cycle.

This underrelaxation also addresses the possibility that $\boldsymbol{x}_c$ might lie outside of the ocean domain. If $\boldsymbol{x}_c$ is masked, then there is no valid velocity to provide via off-grid interpolation, so instead of evaluating (33) $\boldsymbol{x}_c'$ is set to $\boldsymbol{x}_a$ in step 3 of algorithm 3. This combines with the underrelaxation after 10 iterations to reduce the trajectory length until an in-boundary point is found, whereupon iteration resumes normally.

These values for iteration counts and underrelaxation parameters are conservatively specified. In the numerical tests discussed in this work, the vast majority of trajectories converge after one or two iterations, without needing to resort to underrelaxation or trajectory truncation.

## 4.3 Velocity interpolation

The trajectory algorithm requires the off-grid interpolation of velocities at each iteration. In principle, these velocities can be interpolated using the interpolation process of section 3. Doing so would be ideal for ultimate consistency with the final off-grid interpolation, but this process is also computationally expensive. In practice, it is more efficient to evaluate the off-grid velocity field in step 2 of algorithm 3 using trilinear interpolation; doing so causes little change in the numerical test cases in this work.

Trilinear interpolation proceeds with the same order of operations as algorithm (1): velocities are first interpolated in depth to the $(x, y)$ corners of the grid-box at the off-grid level, then along the $x$-direction, and finally along the $y$-direction. Each individual interpolation respects the relevant boundary condition, so for example the $u$-velocity is considered to reflect symmetrically around a boundary in $y$ and $z$ but is constrained to zero at a boundary in $x$.

One complication of linear interpolation, however, is that the velocity points are staggered by half a cell with respect to the physical boundary. In two dimensions, if the tracer point $T(0,0)$ (to use grid-cell coordinates for the tracer grid denoted $T$) is a land point but $T(0,1)$, $T(1,0)$, and $T(1,1)$ are all ocean points, then u-velocity point $U(0,0)$ (denoting the $u$-velocity grid as $U$), halfway between $T(0,0)$ and $T(1,0)$, lies along the boundary. The boundary continues to $U(0, 0.5)$, whereupon $U(0, 0.5)$–$U(0,1)$ lies inside the ocean. This violates a basic assumption of linear interpolation, that the velocity should vary smoothly (and approximately linearly) within the u-cell.

This causes two problems for trajectory computation. The first problem is that after repeated one-dimensional interpolation, the boundary condition is no longer necessarily respected by the interpolated velocity, which can result in a trajectory iteration that "pushes" the departure point into the wall, causing non-convergence. The second problem is that while the interpolation process guarantees continuity of the interpolated field at the cell corners, the boundary conditions can cause large discontinuities along the cell edges, again resulting in a convergence failure. In the above example, the interpolated velocity at $U(+\epsilon, 0.6)$ would be influenced by both $U(0,1)$ and the zero velocity at the physical boundary of $U(0,0)$, but the interpolated velocity at $U(-\epsilon, 0.6)$ would be influenced by $U(0,1)$ and its reflection at a ghost point. These problems are illustrated in the top panel of figure 4.

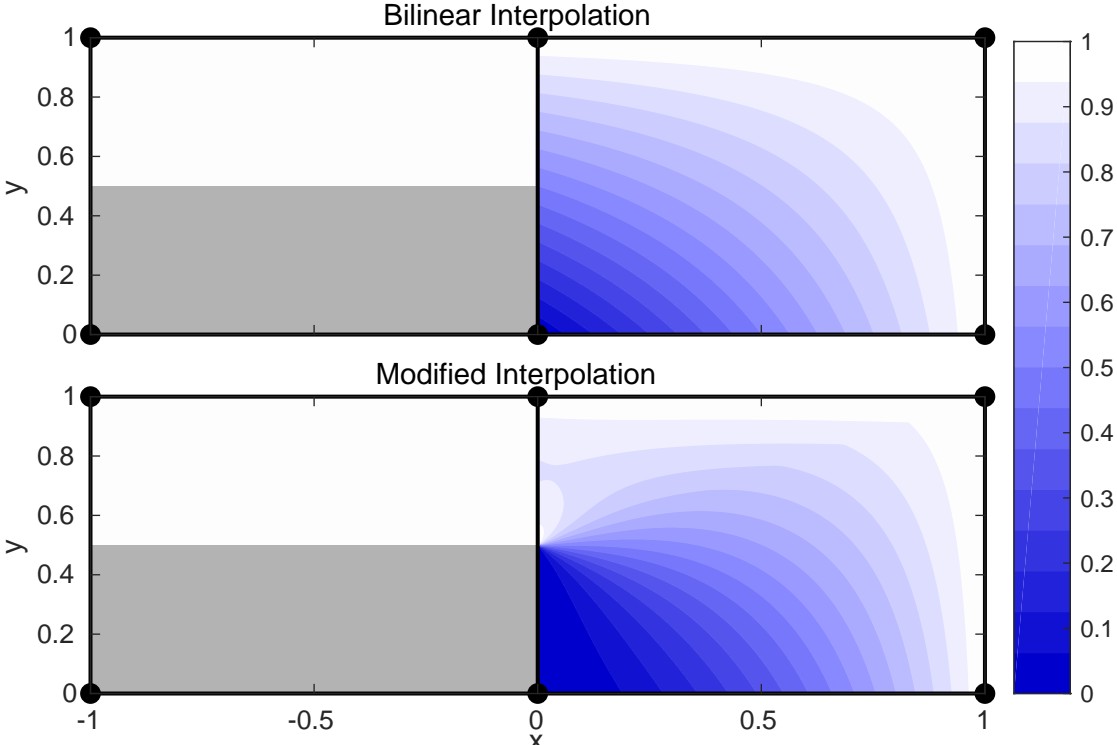

**Figure 4.** Illustration of modified linear interpolation near corners. At top, linear interpolation results in an interpolated velocity field that does not respect the boundary conditions along $0 < y < 0.5$ and a discontinuous interpolated field at $0.5 < y < 1$. At bottom, modifying the linear interpolation with a corner solution results in a field that respects the boundary condition.

The solution to both of these problems is to blend the linearly-interpolated function with a corner singularity solution. A bilinear function is a solution to Laplace's equation ($\nabla^2 f = 0$), so it is reasonable to consider corner solutions that are also solutions to Laplace's equation.

Without loss of generality, consider a grid cell defined by $(x, y) \in [0, 1]^2$, such that there is a solid boundary along ($x = 0, y <$ 0.5) as depicted in figure 4. Treating the boundary as an *infinite* half-plane, with $f(0, y) = 0$ for $y < 0.5$ and $f(0, y) = f(0, 1)$ for $y > 0.5$, the "corner" solution to Laplace's equation is:

$$f_{corner}(x, y) = \frac{f(0, 1)}{2} \left( 1 + \cos\left( \tan^{-1}\left( \frac{y - 0.5}{x} \right) \right) \right), \tag{34}$$

while bilinear interpolation would give:

$$f_{bilinear}(x, y) = (1 - x)y f(1, 0) + x(1 - y)f(0, 1) + xy f(1, 1). \tag{35}$$

These two solutions are blended together, with (34) taking precedence along the solid boundary ($x = 0$ and $0 \leq y \leq 0.5$) and (35) taking precedence along the $x = 1$ and $y = 1$ boundaries of the cell. This gives:

$$f_{blend}(x,y) = \sigma(x,y)f_{bilinear}(x,y) + (1 - \sigma(x,y))f_{corner}(x,y), \tag{36}$$

where $\sigma = \max(1 - x, 2(y - 0.5))$.

   The blended function exactly respects the solid boundary condition, and the discontinuity at the cell edges is significantly
reduced. Blended functions for other configurations of the solid wall are given by applying the appropriate reflections and rotations to (36).

## 5 Results

### 5.1 Flow past an island

To demonstrate the impacts of semi-Lagrangian advection on a simple test case with a lengthened timestep, we first present the
quasi-two dimensional test case of isothermal flow past an interposed island.

   This test case consists of a $280 \times 70 \times 3$ point grid, with grid resolution $\Delta x = \Delta y = 5$m and $\Delta z = 10$m. A $50$m $\times 50$m region ($10 \times 10$ points) is masked as land in the middle of the domain. The inflow boundary condition is set to $u = 0.03$m/s, $v = 0$; this was also imposed throughout the domain as an initial condition. The reference frame was also irrotational, with a Coriolis parameter of 0.

Relevant namelist parameters are given in table 1, with parameters that differ between the control and semi-Lagrangian runs highlighted. The control run used flux-form velocity advection[7] via the QUICKEST scheme (Leonard, 1979, 1991), whereas the semi-Lagrangian run used semi-Lagrangian advection of momentum in flux form as described in sections 3 and 4. To emphasize the dynamical differences between the advection schemes, both test cases were run with no explicit horizontal diffusion of momentum. Vertical mixing terms, largely irrelevant for this quasi-two dimensional case, were set consistently
with the ORCA025 simulations in section 5.2.

   Both series of runs used the implicit free surface formulation (enabled with the compile-time key `key_dynspg_flt`), which damped the large initial surface gravity waves caused by the imposition of the blocking island on the steady-state flow.

   After the initial gravity-wave adjustment, this test case quickly develops a set of recirculating vortices in the lee of the island. Over time these vortices grow in extent and would begin detaching to form a vortex street, but this does not happen before the
8000s end of the simulation. Although there is no explicit horizontal diffusion of momentum in these runs, the flow regime is much more laminar than would be implied by the physical Reynolds number of $1.5 \cdot 10^6$, based on the free-stream velocity, edge-length of the island, and molecular viscosity of water.

   In moving around the box, the flow locally accelerates to a maximum steady velocity of about $0.05 \mathrm{m\,s}^{-1}$, and this maximum velocity is reached in the vicinity of the leading-edge corners of the box. The exact value of this maximum depends on both the
simulation time and the timestep, but our expected pattern holds: the control simulation is stable with a timestep of $64$ seconds,

---
[7]This choice of velocity advection provided the best results for the control run, of the advection models supported in NEMO version 3.1.

| Parameter | Value | Comments |
|---|---|---|
| rdt | *Varies* | Varied from 5s – 160s |
| nitend | *Varies* | Set so that rdt * nitend $= 8000s$ |
| ln_zps | .TRUE. | Enables the z-level coordinate; no partial steps were necessary |
| atfp | 0.1 | Asselin time filter parameter |
| ln_dynvor_een | .FALSE. | Flux-form advection |
| ln_dynvol_qck | .TRUE. | QUICKEST velocity advection (for control run) |
| ahm0 | 0 | Horizontal eddy viscosity for momentum |
| avm0 | 1.2e-4 | Vertical eddy viscosity |
| ln_zdfevd | .TRUE. | Enhanced vertical diffusion |
| avevd | 100 | Vertical coefficient of enhanced diffusion |
| n_evdm | 1 | Apply enhanced vertical diffusion to momentum |
| nn_botfr | 3 | Free slip bottom boundary condition |

**Table 1.** Selected namelist parameters for the test case of section 5.1.

which corresponded to a maximum steady Courant number ($\max(u_{steady})/\Delta x$) above $0.6$ (and a maximum transient CFL of $0.95$), but it is unstable with a timestep of $80$ seconds.

Semi-Lagrangian advection maintains stability for much longer timesteps. Figure 5 shows the free surface height and flow streamlines for $\Delta t$ between 5 and 160 seconds, and only the semi-Lagrangian method remains stable for 80 and 160-second timesteps. For both advection schemes, the longer timestep is associated with a more diffuse flow pattern, with lengthening (and less intense) recirculating vortices in the lee of the island.

This effect is stronger with semi-Lagrangian advection than with Eulerian advection. We attribute this to the nature of the flow at the leading edge of the island. Here, the dominant flow balance is cyclostrophic, where the pressure gradient at these corners balances the local vorticity. The operator splitting method used here treats the advective terms in a frame following the flow, but it can only apply the pressure force at the destination cell. This results in an inconsistency that grows with $\Delta t$, related to the forces in equation (2) being available only at the endpoint of the Lagrangian trajectory – an $O(\Delta t)$ approximation to the integral.

This inconsistency is most evident in the 160-second timestep case (bottom right panel of figure 5), where the maximum steady Courant number of $1.6$ implies that fluid parcels are advected by about three grid cells over the $2\Delta t$ leapfrog step. There, the lowest pressure region at the leading edge of the flow has moved slightly further downstream.

In the full ocean, the geostrophic effect predominates, with a leading-order balance between the pressure gradient and the Coriolis force (planetary vorticity), so we expect this issue to be less pronounced.

## 5.2 Global forced runs

To evaluate semi-Lagrangian advection in a more realistic forecasting setting, we conducted a preliminary series of free runs of the NEMO-OPA model. The runs consisted of:

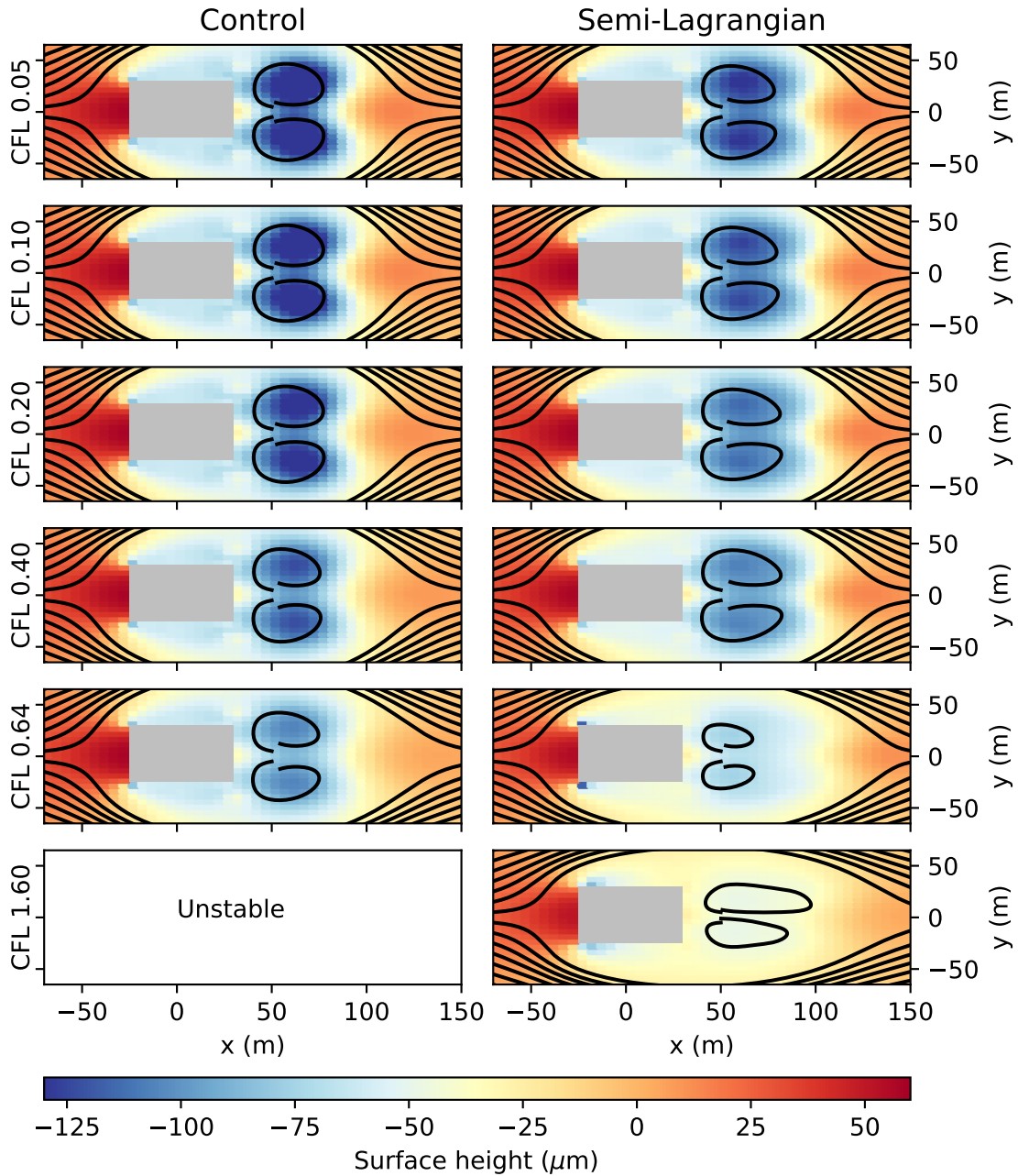

**Figure 5.** Free surface height and streamlines for the test case of section 5.1, after 8000s for $\Delta t = 5, 10, 20, 40, 64,$ and $160$ seconds (top to bottom, with the approximate Courant number listed). Results for the Eulerian advection scheme are at left, and results for the semi-Lagrangian advection of momentum are at right. As the timestep increases both advection schemes show more diffuse behaviour, however the semi-Lagrangian advection scheme remains stable to $\Delta t = 160$s whereas the Eulerian scheme becomes unstable after $\Delta t = 64$s.

– A control run, based on the configuration of Environment and Climate Change Canada's $1/4\circ$ nominal-resolution Global Ice/Ocean Prediction System (GIOPS) (Smith et al., 2016) with a 10-minute model timestep[8]. Tracers were advected with the model's tracer variance dissipation scheme (Lévy et al., 2001), and momentums were advected in vector form with the model's energy and enstrophy conserving scheme[9] (Arakawa and Lamb, 1981),

– A "semi-Lagrangian tracer" run, where momentum was advected as in the control scheme and the semi-Lagrangian advection described in this work was used for advection of salinity and temperature. Additionally, this run disabled horizontal diffusion of salinity and temperature, and

– A "semi-Lagrangian momentum and tracer" run, where momentum as well is advected with the semi-Lagrangian scheme. The configuration was otherwise the same as the semi-Lagrangian tracer run, save for a 15-minute model
timestep.

The runs were all initialized at October 1, 2001 on the ORCA025 grid. The ocean was at rest, and temperature and salinity were given by the 2011 World Ocean Atlas climatology (Locarnini et al., 2013; Zweng et al., 2013). Atmospheric forcing was provided at one-hour intervals from Environment and Climate Change Canada's $1/4°$ global atmospheric reforecast, and sea ice was modeled via coupling with version 4.0 of the CICE model (Hunke and Dukowicz, 1997), with dynamically active
(moving) ice. Selected namelist parameters are provided in table 2.

As with section 5.1, the test cases used NEMO-OPA's linear free surface with a time-implicit solver, and tidal forcing was not present in these configurations. In a typical timestep, the vast majority of semi-Lagrangian trajectories converged in one iteration (mean 1.004 over the "semi-Lagrangian tracer" run). A very small minority of cells required an extended number of iterations or underrelaxation as described in section 4.2, but this did not affect the overall trajectory-calculation performance
because convergence was measured (and iterations limited) on a per-cell basis.

Each run continued through late 2009. For reasons of space efficiency, we recorded the two-dimensional sea surface height, temperature, and salinity fields for each model day, and we preserved every fifth daily-mean, three-dimensional output of temperature, salinity, and horizontal ocean velocity.

For short and medium-term forecasts, the operational coupled forecasting systems at CMC are constrained by observations
and periodic re-initialization. With a focus on this forecasting horizon the objective with these long free-runs was:

– To provide a test of model stability with semi-Lagrangian advection, in terms of both avoiding crashes and providing plausible ocean fields;

---

[8]This timestep is shorter than other commonly-used ORCA025 configurations, such as in the ocean reanalysis of Ferry et al. (2016). This shorter timestep is required to stabilize the coupling of ocean/ice stress with the CICE model, where following Roy et al. (2015) the ice/ocean drag coefficient is larger than typically considered. We chose to maintain this configuration and coupling approach to provide for the cleanest like-for-like comparisons against the operational configuration

[9]For compatibility with the operational model, as run in this work the scheme did not include the "fix" for the Hollingsworth instability(Hollingsworth et al., 1983) reported in Ducousso et al. (2017). This instability is more prominent at higher resolutions, and we do not believe it meaningfully impacted the results as presented in this section.

| Parameter | Value | Comments |
|---|---|---|
| *Parameters common to all runs* | | |
| atfp | 0.1 | Asselin time filter parameter |
| ln_zps | .TRUE. | Z-level vertical coordinate with partial (cut) cells |
| e3zps_min | 25 | Absolute minimum thickness of a cut cell |
| e3zps_rat | 0.2 | Relative minimum thickness of a cut cell |
| shlat | 0 | Free-slip lateral momentum boundary condition |
| nn_botfr | 2 | Nonlinear bottom friction |
| nn_bfro2 | 1e-3 | Nonlinear bottom friction coefficient |
| nn_bfeb2 | 2.5e-3 | Background turbulent kinetic energy coefficient |
| ngeo_flux | 0 | No bottom temperature geothermal heat flux |
| ln_dynhpg_imp | .TRUE. | Semi-implicit computation of the hydrostatic pressure gradient |
| ln_dynldf_bilap | .TRUE. | Bi-Laplacian hyperdiffusion of momentum |
| ln_dynldf_hor | .TRUE. | …acting in the horizontal direction |
| ahm0 | -3e11 | Momentum hyperviscosity coefficients |
| nsolv | 2 | Use the successive over-relaxation (SOR) free-surface solver |
| nsol_arp | 0 | …with an absolute-tolerance stopping condition |
| nn_sstr | 0 | No sea surface temperature damping |
| nn_sssr | 0 | No sea surface salinity damping |
| ndmp | 0 | No temperature or salinity damping in the water column |
| *Parameters for the control run* | | |
| rdt | 600 | Model timestep |
| ln_traadv_tvd | .TRUE. | Tracer variance dissipation (TVD) tracer advection scheme |
| ln_traldf_lap | .TRUE. | Laplacian diffusion for the tracer |
| ln_traldf_iso | .TRUE. | …acting in the iso-neutral direction |
| aht0 | 300 | Horizontal tracer diffusion coefficient |
| ln_dynadv_vec | .TRUE. | Vector form of the momentum advection operator |
| ln_dynvor_een | .TRUE. | …using the energy and enstropy conserving scheme |
| resmax | 1e-10 | Absolute residual tolerance for the SOR free-surface solver |
| *Parameters for the semi-Lagrangian tracer run* | | |
| rdt | 600 | Model timestep |
| ln_traldf_lap | .FALSE. | *No* explicit horizontal tracer diffusion |
| ln_dynadv_vec | .TRUE. | Vector form of the momentum advection operator |
| ln_dynvor_een | .TRUE. | …using the energy and enstropy conserving scheme |
| resmax | 1e-11 | Absolute residual tolerance for the SOR free-surface solver |
| *Parameters for the semi-Lagrangian momentum and tracer run* | | |
| rdt | 900 | Model timestep |
| ln_traldf_lap | .FALSE. | *No* explicit horizontal tracer diffusion |
| ln_dynadv_vec | .FALSE. | Flux form of the momentum advection operator |
| resmax | 1e-11 | Absolute residual tolerance for the SOR free-surface solver |

**Table 2.** Selected dynamical and numerical namelist parameters for the test cases of section 5.2.

- To check for any large-scale conservation errors, which might be difficult to correct given the sparsity of observation data for the deep ocean, and

- To note any qualitative improvement or deterioration in the effective resolution of the model.

This first goal of model stability was met in part by the successful completion of these runs. Use of semi-Lagrangian advection for both tracers and momentum allowed us to increase the effective timestep from 10 minutes (with typical maximum Courant number[10] of 0.2, found in the vertical direction) to 15 minutes (Courant number 0.3). Further increases led to instability and model crashes not from the advection component, but from the ice model. In this version of the model, the ocean/ice stress is coupled in a time-explicit way between the water and ice components. Concurrent work towards a time-implicit coupling has given encouraging preliminary results on further timestep increases.

The use of semi-Lagrangian advection also gives global flows qualitatively similar to the control run, and average transports in the Atlantic overturning circulation and Circumpolar current are comparable between the control and semi-Lagrangian runs (figure 6). The semi-Lagrangian runs appear to result in a slightly weaker overturning circulation and a slightly stronger circumpolar current than the control run, but these results may not be robust to re-tuned physical paramterizations. Using semi-Lagrangian advection for the velocity components results in a significant decrease to overall ocean kinetic energy (figure 7), both during and after the spin-up period.

The cause of this energy disparity is under investigation, but we believe the most likely cause is the application of slope-limiting to the $u$ and $v$ fields independently. Future work will focus on taking a more nuanced approach to filtering, but this effect may not be very significant in a shorter-term forecast setting with frequent re-initializations from an analysis.

The second goal of global conservation was met. Although semi-Lagrangian advection does not guarantee conservation of tracers, the impact on the global balance of temperature and salinity was small. Figure 8 shows the evolution of ocean-average temperature and salinity over time in these runs, and the effect of non-conservation attributable to the semi-Lagrangian advection of tracers is comparable to the magnitude of uncertainty in the global balance of atmospheric forcing – the imbalance seen in the control run. Each case saw an overall temperature drift of about 0.04 K over the simulated period, with the semi-Lagrangian cases having a slight warming trend against the control run's slight cooling trend, and all three runs showed a very small increase in ocean average salinity, by about 0.01 PSU.

The temperature change versus depth over the simulated period is shown in figure 9. Both the control and semi-Lagrangian runs showed a warming trend in the surface layers, but the semi-Lagrangian runs showed temperature stability in fluid layers below 1000 meters depth whereas the control run showed a cooling trend in these waters.

Despite the energy shortfall with semi-Lagrangian advection of momentum, we see tentative signs that the method increases the model's effective resolution. Figure 10 shows one particular sea surface temperature realization, from the 31 December 2005 of each test case, along with the magnitude of the temperature gradient. The large-scale flows are similar between the control and semi-Lagrangian runs (and most similar between the control and semi-Lagrangian tracer run), but the semi-Lagrangian runs have noticeably stronger gradients in the sea surface temperature, in patterns that resemble smaller-scale eddies.

---

[10]Defining the Courant numbers in each direction as $\max(|u|)/\texttt{e1u}$, $\max(|v|)/\texttt{e2v}$, and $\max(|w|)/\texttt{e3w}$ respectively.

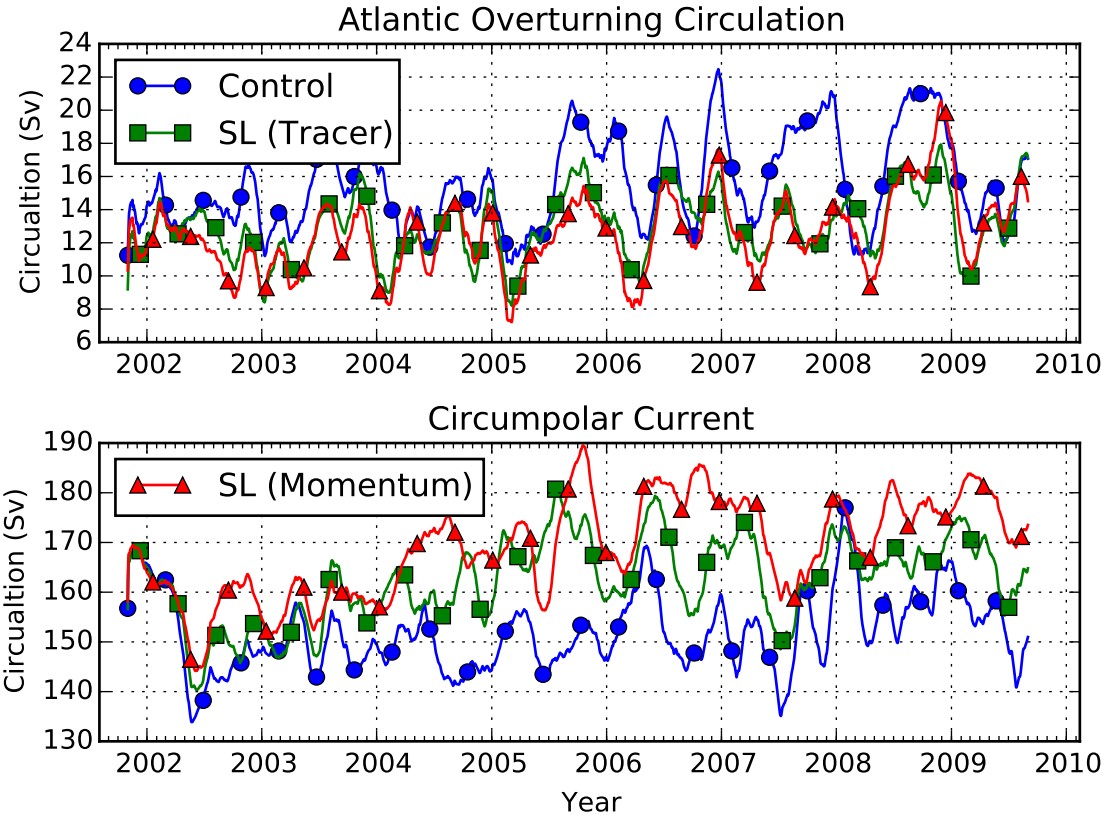

**Figure 6.** 61-day mean transports for the Atlantic overturning circulation (top; net northward flux above 1000m depth at $27.25°$ north latitude in the Atlantic Ocean) and Antarctic circumpolar current (bottom; net eastward flux at $67.75°$ west longitude in the Drake Passage) over time for the test cases of section 5.2

## 6 Conclusions and further work

This work has derived a semi-Lagrangian advection scheme for the NEMO-OPA model. After advecting a tracer or momentum field along estimated fluid parcel trajectories, it calculates a time-trend to provide to the remainder of the model; in this way the semi-Lagrangian scheme serves as a drop-in replacement for other tracer and (flux-form) momentum schemes.

The development of this advection module relied on several new or newly-applied algorithms that might be relevant to other ocean models or other domains:

–  The "semi-Lagrangian trend" form of equation (7) might be useful in other models when researchers wish to implement semi-Lagrangian advection after the fact, without disrupting the calculation of other forcing terms.

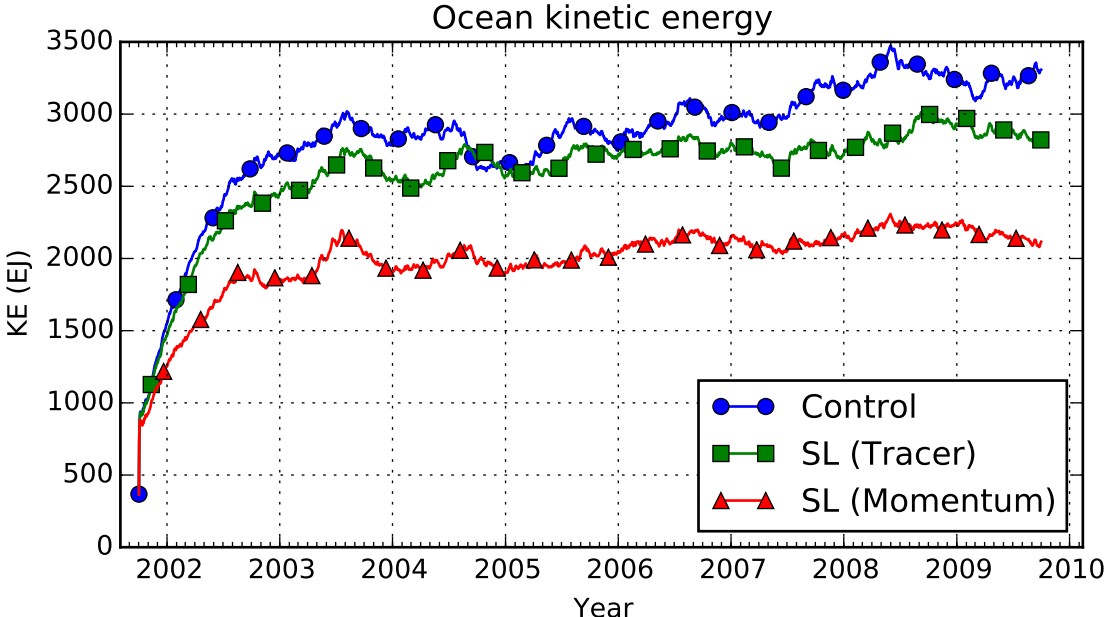

**Figure 7.** Total ocean horizontal kinetic energy (EJ) over time for the test cases of section 5.2. All of the test cases generally reproduce the monthly to yearly variability of kinetic energy, but the use of semi-Lagrangian momentum advection results in significantly lower total kinetic energy.

- The Hermite interpolation form in section 3, especially combined with the $C^1$-continuous estimate of the vertical derivative in section 3.2 might find application in other domains where, as in the ocean, one dimension (the vertical) is more oscillatory than others.

- The exponential integration of trajectories in (4) may be useful in other applications that feature strong accelerations over trajectories. In particular, it forbids trajectory-crossing in one dimensional flows, and here that property ensures that trajectories remain inside the ocean domain.

- The "corner solution" treatment of velocity for trajectory calculations near corners might find use in other applications with staggered velocity components.

Overall, we find that the semi-Lagrangian method is effective at extending the realizable timestep in the NEMO-OPA model. In the simple domain of section 5.1, this method resulted in a stable simulation with advective Courant numbers in excess of 1. Although we only extended the timestep from 10 to 15 minutes for the semi-Lagrangian momentum run in section 5.2, this limitation was imposed by the ice model. Disabling ice dynamics allowed us to increase the timestep to 30 minutes, but this would have made the results incomparable with those of the control and semi-Lagrangian tracer runs. Preliminary work with the CICE sea model and implicit coupling of the ice-ocean stress seems to allow us to relax the ice-related timestep restriction.

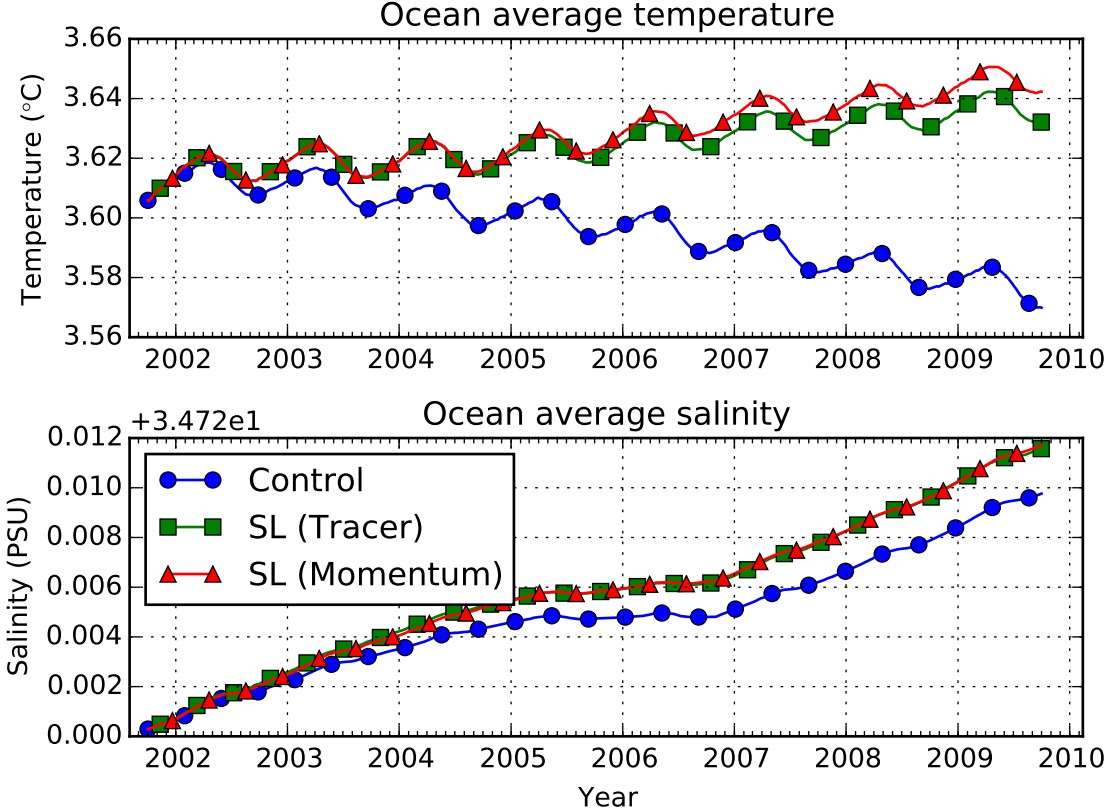

**Figure 8.** Ocean average temperature (top, °C) and salinity (bottom, PSU) over time for the test cases of section 5.2. Although conservation is not guaranteed by semi-Lagrangian advection, long-term trends are similar between the semi-Lagrangian runs and the control run.

**Performance and implementation**

In spite of this increased timestep, the semi-Lagrangian method by itself does not yet improve overall computational perfor-
mance. The semi-Lagrangian momentum and tracer run of section 5.2 took approximately one hour of computational time
per five days of simulated time, using 128 Intel Xeon E5530 processors at 2.4GHz. With a 10-minute timestep, the semi-
Lagrangian tracer run took approximately 50 minutes for the same five days of simulated time, whereas the control run took
just 30 minutes. We expect these results to improve with further numerical optimization work. In particular, we did not take
great care to ensure that loops were vectorized where possible, and it is much more difficult for compilers to automatically
vectorize the point-by-point semi-Lagrangian computations compared to volume flux calculations in the traditional advection
schemes.

About one-third of the additional computational cost comes from trajectory iterations, and the remainder comes from the
cubic interpolation. This suggests that the relative cost of semi-Lagrangian advection will be lower than presented here if tra-

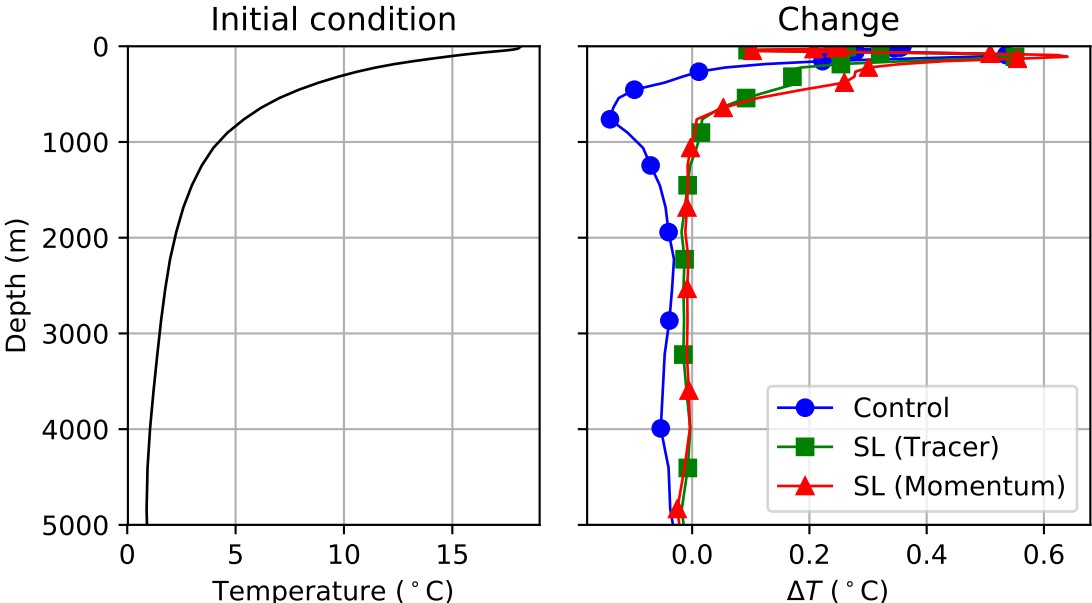

**Figure 9.** Initial ocean-average temperature profile (left, °C) versus depth and change at the end of the simulated period (right, 4 October 2009) for the test cases of section 5.2. Both semi-Lagrangian runs show temperature stability in deeper waters, whereas the control run shows a small cooling.

jectories can be reused for multiple tracer species (such as biogeochemical constituents). Additionally, it suggests that a further
optimization may be to re-use tracer trajectories for momentum advection, at least away from the boundaries where interpolating (staggering) trajectories might be reasonable. It seems unlikely that optimization will reduce the per-timestep penalty to the 20% value seen by Ritchie et al. (1995) for an atmospheric – model owing to the lack of three-dimensional implicit equations and expensive physical parameterizations elsewhere in NEMO-0PA – but we are hopeful that semi-Lagrangian advection will nonetheless improve overall system performance.

The parallel (MPI) implmenetation of this algorithm was straightforward. With the relatively modest increase in Courant number for the cases in this work, we simply needed to increase the inter-processor lateral halo (parameters `jpreci` and `jprecj`) to three points, which was sufficient to allow a fluid parcel arriving at a processor's edge to apply the full interpolating stencil for the Courant numbers reached in the presented simulations. This increase in halo size was small compared to the processor tile size of about $50 \times 260$ grid points for the runs in section 5.2. Extending this to support very large horizontal
Courant numbers, however (if another solution could be found to stabilize baroclinic waves) would require either prohibitively large halo sizes or additional interprocessor communication to track fluid parcels that cross MPI boundaries.

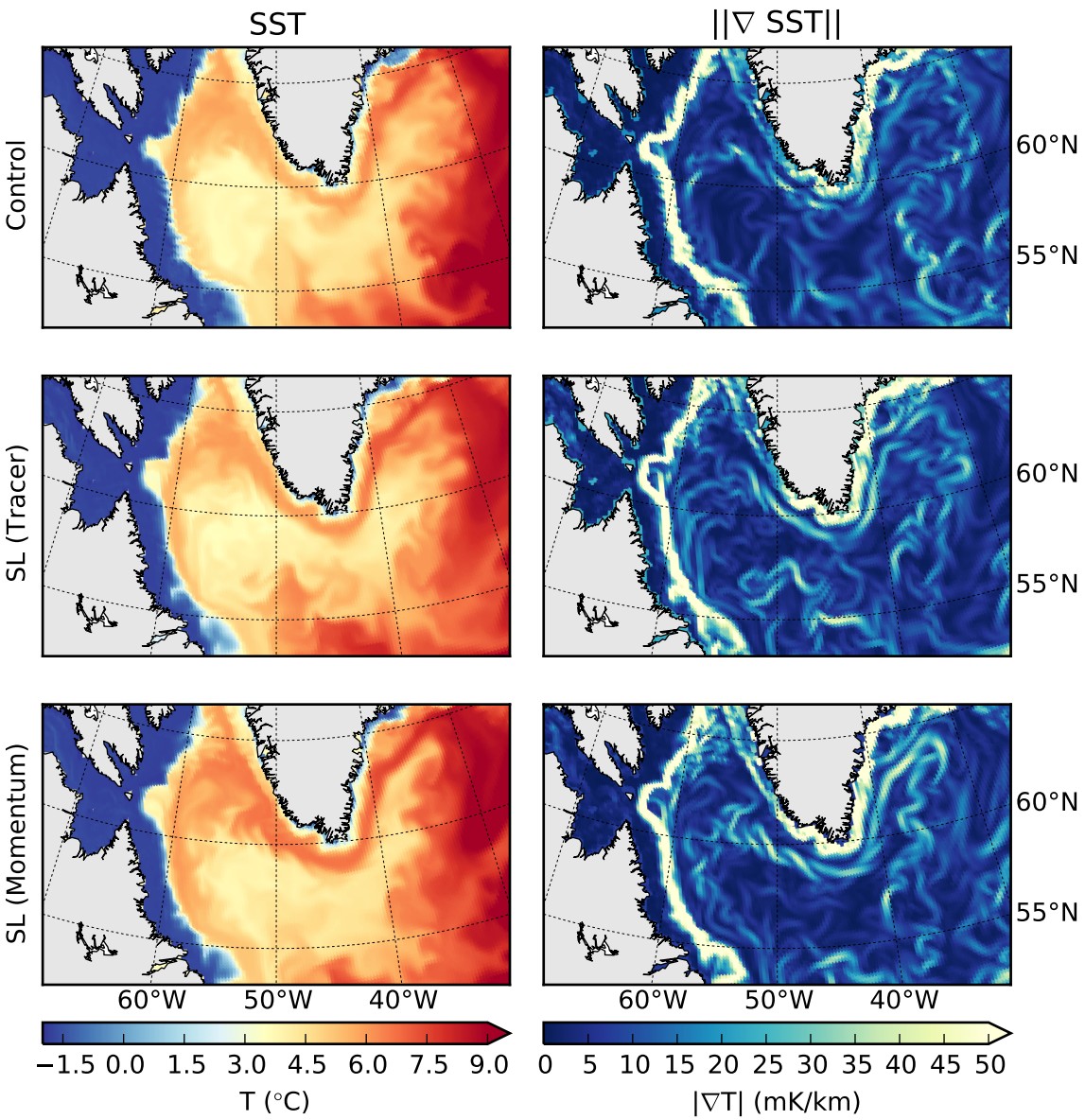

**Figure 10.** Sea surface temperature (left) and the magnitude of its gradient (right) for the control (top), semi-Lagrangian tracer (middle), and semi-Lagrangian momentum and tracer (bottom) test cases of section 5.2, for 31 December 2005 in the Labrador Sea. Although the large-scale flows are similar, the runs with semi-Lagrangian advection of tracers have noticeably more fine-scale variability.

**Qualitative comments on results**

Although the semi-Lagrangian method does not guarantee tracer conservation, we see no evidence that its implementation here leads to a degradation relevant in a weekly to seasonal forecast setting. In particular, the deep-water temperature stability shown in figure 9 is an encouraging sign that semi-Lagrangian advection will preserve the deep-water structure that is weakly constrained by data. Even small imbalances, however, might become significant over the decade-to-century timescales of climate simulations. Further work will be necessary to characterize this method before we can safely recommend semi-Lagrangian advection in such settings.

For the test cases in section 5.2, semi-Lagrangian advection of tracers appears to slightly increase the effective resolution of the model. However, this effect is much more mixed when momentum is also advected with the semi-Lagrangian method, in part because the underlying currents differ. Both of these differences will be the subject of future study, with the specific intention of assessing these effects in the setting of shorter-term forecasts. We speculate that the overall loss of kinetic energy with semi-Lagrangian advection of momentum is attributable to the use of the slope limiter: limiting each component of velocity separately may be causing unrealistic diffusion of smaller-scale structures in the presence of background vorticity. We hope to address this issue with more selective limiting.

**Future development**

Finally, the development in this paper implicitly assumes that the coordinate system is static with time. This is not the case in NEMO-OPA when using its nonlinear free surface option, which necessarily implies time-varying vertical levels. Adapting the semi-Lagrangian method to this more general coordinate system will be a focus of future work, which will be required to apply this advection scheme to higher-resolution domains that require tide-permitting simulations.

Additionally, future versions of NEMO intend to move to a third-order Runge-Kutta time-stepping algorithm (Shu and Osher, 1997), which constructs a full timestep as a linear combination of forward Euler steps. We expect that the semi-Lagrangian "advective trend" of (7) can be adapted to this framework in a straightforward manner by basing the calculated trend on the current-step values of tracers and velocities, but the adaptation may require care to preserve the higher-order temporal accuracy of the overall scheme.

*Code availability.* The modified NEMO (CeCILL license, version 2.0) code along with scripts and data used in this paper are available under Subich et al. (2020). The modified CICE 4.0 used in section 5.2 is not redistributable under a free license, but it has been made available for the topical editors and anonymous reviewers.

*Author contributions.* All authors were responsible for the concept. CS, PP, and FD were responsible for the necessary software development. CS contributed to the original draft preparation, and all authors contributed to review and editing.

*Competing interests.* The authors declare that they have no competing interests.

*Acknowledgements.* The authors would like to acknowledge Francois Roy of Environment Canada, who provided considerable technical support, especially with the forcing runs of section 5.2, and Jerome Chanut of Mercator Ocean, who provided helpful comments on a draft of this paper. The authors also are grateful for the peer review provided by Florian Lemarie and Mike Bell, which greatly improved this paper
from its original submission.

    The authors also owe a debt of inspiration to the semi-Lagrangian dynamical core of MC2 (Thomas et al., 1998).

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
