# Peer review of "Development of a semi-Lagrangian advection scheme for the NEMO ocean model (3.1)"

_Geoscientific Model Development, 2020_

## Short Comment (SC1) · 23 Apr 2020

Dear authors,

in my role as Executive editor of GMD, I would like to bring to your attention our Editorial version 1.2:

https://www.geosci-model-dev.net/12/2215/2019/

This highlights some requirements of papers published in GMD, which is also available on the GMD website in the 'Manuscript Types' section:

http://www.geoscientific-model-development.net/submission/manuscript_types.html

In particular, please note that for your paper, the following requirement has not been

met in the Discussions paper:

- "The main paper must give the model name and version number (or other unique identifier) in the title."

Please add a version number for NEMO in the title upon your revised submission to GMD.

Yours,

Astrid Kerkweg

---

## Referee Comment (RC1) · Florian Lemarie (Referee) · 4 May 2020

This paper describes the implementation of a semi-Lagrangian (SL) advection scheme in the NEMO oceanic model. Sec. 2 discusses the fundamental difference between the Eulerian and SL formulation of advection. It is also shown how a SL advection fits within the existing NEMO framework. Such advection algorithm is made of two specific ingredients: (1) a backward integration to determine the departure points of trajectories (2) an interpolation from the on-grid values to the off-grid departure points found in step (1). The interpolation is described in Sec. 3 while the back trajectory computation is discussed in Sec. 4. Sec. 3 also shows a comparison between the proposed SL algorithm and a standard Eulerian scheme for a simple two-dimensional advection problem with prescribed velocity to check numerically the proper implementation of the

interpolation step. Finally in Sec. 5, two additional numerical tests are presented. A first semi-idealized test of a flow past an Island is implemented to test the adequacy of the proposed algorithm to find departure points for the trajectories. Then, results from realistic global simulations at $1/4$-degree resolution are shown with a systematic comparison between the SL approach and the Eulerian one. In the conclusion, the authors provide the remaining steps toward a more robust and efficient implementation of their SL algorithm in the NEMO model.

The paper is generally well written and the underlying motivations for the study are clearly stated. The authors try to tackle a very challenging task for numerous reasons: (1) the NEMO code has been fundamentally thought and built to work with Eulerian advection schemes (2) in the oceanic community SL approaches are not very popular because they do not conserve the domain integral of a tracer, they are thought to be very dissipative, and also because there is a general belief that the time-step of oceanic models is mostly constrained by internal gravity waves propagation (3) besides the numerical issues described in the manuscript, the actual code implementation of a SL algorithm in the context of parallel computing is a arduous task to guarantee both an acceptable time-to-solution and scalability. It is always very instructive to explore alternatives to challenge approaches well established for many years. Even if the paper does not conclude to an indisputable success of the proposed method in the context of global realistic applications, **I recommend publication of the manuscript provided the revisions mentioned below. The paper clearly fills a gap in the oceanic modelling community literature and fits perfectly in the GMD NEMO special issue. Moreover, I particularly liked the honesty of the authors because they do not try to oversell their method despite the significant amount of work needed to get it accomplished (the authors deserve the greatest respect for bringing their work to completion). The manuscript follows a scientifically sound reasoning with systematic testing of the different blocks of the proposed method**. I think a bit of work is necessary in the introduction and in Sec. 2 to put the reader in the best possible conditions for the rest of the paper. Moreover, I would expect more discussion about

the performance issues in Sec. 5 and 6 and compelling arguments for not accounting for the Robert-Asselin filter in your analysis.

Please see below my comments and suggestions.

**Major remarks and suggestions :**

- I initially had a negative feeling about the paper during my first read because I was asking myself several questions (only partially answered in the conclusion) which diverted my attention. For example, the fact that the proposed method is not conservative and is implemented only in the linear free-surface case (i.e. when the vertical grid does not move with time) should be mentioned from the introduction. Another question not tackled in the paper is about the interference between the Robert-Asselin filtering and the SL algorithm for nonlinear simulations. I don't think it is so obvious that the time filtering does not impact the general rationale developed in Sec. 2 (see my comment below)

- **Title**: since the paper does not describe a completely finalized work applicable whatever the numerical options available in NEMO, I would suggest a title like: "*Semi-Lagrangian advection in the NEMO ocean model: implementation and first experiments*". From the conclusions of the present study we can anticipate a subsequent publication with more mature results.

- **Introduction and Sec. 2:** some work is needed to be more pedagogue considering that a typical reader would be either familiar with the SL methods but not with oceanic modeling (e.g. a typical reader from the atmospheric community) or the other way around (i.e. a typical reader from the oceanic community). In

my sense, what makes your study challenging is that you have to deal with a problem under constraints. You already explain the operational constraints but I would also explain more clearly two other levels of constraints to unambiguously justify that you have to deal with problems not already tackled by the atmospheric community

– **Specificity of the oceanic context with consequences on the design of SL schemes:**

→ *Geometric problems due to the coastlines* ⇒ impact on trajectory calculation

→ *Stiffness of the problem*. In particular, you take for granted that the time-step of global configurations is constrained by the advective CFL, without supporting reference. From my own experience, it has been difficult to convince ocean modelers that the time-step of global simulations could be mostly limited by advection rather than internal gravity waves (IGW) propagation (I usually do not like to suggest citing my own work but you may use Lemarié et al. (2015) and at a lesser extent Shchepetkin (2015) to further motivate the fact that the time-step is indeed constrained by the advective CFL at $1/4$-degree resolution and higher). After all, your results shown in Sec. 5 and some developments done recently in NEMO (see below) confirm this is indeed the case for ORCA025 configurations.

→ *Mode-splitting for the treatment of external gravity waves* (it would be the occasion to make clear that the computational grid is considered fixed in your study). There is no such thing as mode-splitting between a 2D barotropic and a 3D baroclinic mode in the atmospheric context. The overwhelming majority of NEMO applications are done with $z^\star$ coordinate (i.e. the grid follows the barotropic motions) and ALE

coordinates will be increasingly used in the future.

– **Specificity of the NEMO context with consequences on the design of SL schemes:**

  → *Structured ORCA grid* ⇒ small-cell problem. You seem to consider that the small-cell problem by itself justifies that the time-step is constrained by advection. However small cells also penalize the stability limit associated with IGWs.

  → *Time stepping algorithm* ⇒ Leapfrog (with Robert-Asselin filtering in nonlinear settings)

  → *Grid-staggering* ⇒ C-grid less natural and efficient than A-grid for SL methods

– You could also make it clear in the introduction that your approach considers a finite-difference interpretation of model variables (in NEMO, tracers are often interpreted in a finite-volume sense).

– In subsection 1.2, you don't give any detail on the existing work regarding the interpolation step. The conservation issue could be mentioned in this subsection and maybe you could motivate why you do not consider conservative SL approaches like the cell-integrated semi-Lagrangian (CISL) schemes or a Lin & Rood type methodology (e.g. Lauritzen, 2007).

– In Sec. 2, the notations could be improved at least to make it more explicit which quantities are on-grid and off-grid and which time-level should be considered. For example in eq. (1), $f^{t_0+\Delta t}(\boldsymbol{x})$ is computed knowing $f^{t_0-\Delta t}(\boldsymbol{x})$ at the same grid point such that $\boldsymbol{x}$ is constant with time and corresponds to the standard NEMO fixed computational grid. I would suggest to give a

specific notation for this fixed reference grid (e.g. $x_\text{ref}$). If I understand well, in Eq. (3) $x(t_0 + \Delta t)$ exactly corresponds to $x_\text{ref}$ and what is noted $x(t)$ (it should probably be $x(t_0)$ ?) corresponds to the departure point $x_d(t_0)$ with $x_d(t_0) \neq x_\text{ref}$. In this case it would make very clear that $\text{RHSL}_f^{t_0}(x_d)$ has to be evaluated off-grid in the SISL approach. Some RHSX terms have a superscript for time and some others don't which complicates the proper understanding. Personally I think that something around the following lines to reformulate equations (1) to (6) would be much easier to understand for a reader:

$$(1) \Rightarrow \quad f^{t_0+\Delta t}(\mathbf{x_\text{ref}}) = f^{t_0-\Delta t}(\mathbf{x_\text{ref}}) + 2\Delta t \left\{ \text{ADV}_f^{t_0}(\mathbf{x_\text{ref}}) + \text{RHS}^{t_0 \pm \Delta t; t_0}(\mathbf{x_\text{ref}}) \right\}$$

$$(3) \Rightarrow \quad f^{t_0+\Delta t}(\mathbf{x_\text{ref}}) = f^{t_0}(\mathbf{x}_d^{t_0}) + \frac{\Delta t}{2} \left\{ \text{RHS}^{t_0+\Delta t}(\mathbf{x_\text{ref}}) + \text{RHS}^{t_0}(\mathbf{x}_d^{t_0}) \right\}$$

$$(4) \Rightarrow \quad f^{t_0+\Delta t}(\mathbf{x_\text{ref}}) = f^{t_0-\Delta t}(\mathbf{x}_d^{t_0-\Delta t})$$

$$(5) \Rightarrow \quad f^{t_0+\Delta t}(\mathbf{x_\text{ref}}) = f^{t_0-\Delta t}(\mathbf{x_\text{ref}}) + 2\Delta t \, \text{ADV}_f^{t_0}(\mathbf{x_\text{ref}}) = f^{t_0-\Delta t}(\mathbf{x}_d^{t_0-\Delta t})$$

$$(6) \Rightarrow \quad \text{ADV}_f^{t_0}(\mathbf{x_\text{ref}}) = \frac{1}{2\Delta t} \left( f^{t_0-\Delta t}(\mathbf{x}_d^{t_0-\Delta t}) - f^{t_0-\Delta t}(\mathbf{x_\text{ref}}) \right)$$

At this point, I reiterate my question regarding the impact of the Robert-Asselin (RA) filtering. For example, looking at equation (A.5) in Shchepetkin and McWilliams (2005) it can be found that your equation (1) with RA filter and $\text{RHS}^{t_0 \pm \Delta t; t_0}(\mathbf{x_\text{ref}}) = 0$ would be

$$f^{t_0+\Delta t}(\mathbf{x_\text{ref}}) = (1-2\nu) f^{t_0-\Delta t}(\mathbf{x_\text{ref}}) + 2\nu f^{t_0}(\mathbf{x_\text{ref}}) + 2\Delta t \left\{ \text{ADV}_f^{t_0}(\mathbf{x_\text{ref}}) + \nu \text{ADV}_f^{t_0-\Delta t}(\mathbf{x_\text{ref}}) \right\}$$

with $\nu$ the RA filter parameter ($\nu = \text{atfp} = 0.1$ in your simulations). Please explain how you can reconcile it with your equation (6) ?

Interactive
comment
- **Stability constraints for Eulerian schemes:** throughout the paper the exact form of the stability constraint for Eulerian schemes is fuzzy. In Sec. 3.3 (l. 327) the exact constraint for the Leapfrog (LF) with C2 discretized on a C-grid should be:

$$\Delta t \left( \frac{\max(u_{i+1/2,j},0) - \min(u_{i-1/2,j},0)}{\Delta x_{i,j}} + \frac{\max(w_{i,j+1/2},0) - \min(w_{i,j-1/2},0)}{\Delta z_{i,j}} \right) \leq 1$$

Note that in 1D the LF-C2 scheme is exact for a Courant number equals to $1$. For this reason it could be interesting to add a third case with a maximum Courant number around $0.95$ to see how the two approaches are robust to changes in time-step value.

In Sec. 5.1 you consider the QUICKEST scheme which is a coupled space-time approach implemented via a directional splitting[1], meaning that the associated stability constraint is

$$\Delta t \max \left\{ \frac{\max(u_{i+1/2,j},0) - \min(u_{i-1/2,j},0)}{\Delta x_{i,j}}, \frac{\max(w_{i,j+1/2},0) - \min(w_{i,j-1/2},0)}{\Delta z_{i,j}} \right\} \leq 1$$

which is less restrictive than the one of LF-C2 (in the absence of directional splitting, the stability constraint for QUICKEST in 2D is much more difficult to predict). You should also define how you compute the Courant number in a discrete sense in the SL case on a C-grid. Moreover, is your scheme subject to a Lipschitz criterion ?

- **Courant vs CFL number:** throughout the paper there is a confusion between the Courant number and the CFL number. Please correct it. For example in the
* * *
[1]I have been aware of early implementations of the QUICKEST scheme in NEMO without directional splitting. This is just wrong because the resulting multi-dimensional scheme would be unconditionally unstable in the absence of any external source of dissipation. Hopefully you consider a properly implemented version of it.

[Figure]

caption of Fig. 3 you should not talk about CFL number here but about maximum Courant number (the CFL number for LF-C2 is $1$ whatever the time-step value). Same thing in Fig. 5 where you should not talk about CFL but maximum Courant number (again the CFL is constant and is equal to $1$ for the Control run).

- **Sec. 5.2 and 6:**

  - It is striking to see that the time-step is $\Delta t = 10$ min for your control run with NEMO-CICE whereas with NEMO-LIM3 the standard time-step for ORCA025 is $\Delta t = 20$ min. It seems that there is clearly room for improvement in your ocean/sea-ice coupling.

  - Interestingly, the methodology described in Shchepetkin (2015) to remove the CFL constraint from vertical advection has been recently implemented in NEMO and allowed to further increase the time-step of ORCA025 up to $\Delta t = 30$ min for a marginal increase of the computational cost per time-step. This result is consistent with your remark l. 583-584.

  - The fact that you can increase from $\Delta t = 10$ min to $\Delta t = 15$ min with your SL approach suggests that the advective CFL for ORCA025 with Eulerian advection should lead to $\Delta t = 10$ min (independently from the sea-ice coupling which apparently sets a limit at $\Delta t = 15$ min). However I don't quite understand why your time-step is limited at $\Delta t = 10$ min whereas with NEMO-LIM3 it can be increased to $\Delta t = 20$ min with basically the same advection schemes. You should be able to go at least to $\Delta t = 15$ min for your control run.

  - Given the elapsed time to solution you report in Sec. 6 we can easily find that the computational cost per time-step is 3 times larger with the SL
approach compared to the Eulerian approach (it could be said explicitly in the paper). To be competitive you thus need to be able to take a time-step three times larger. Could you tell what is the contribution of the trajectory computation vs the interpolation step in the computational overhead ? If the overhead is due to the trajectory computation (which has to be done once whatever the number of tracers) it means that your method could gain efficiency in case it is needed to integrate a large number of tracers provided that you can guarantee positivity.

– For the ORCA025 simulations discussed in Sec. 5.2, what is the maximum number of iterations for the trajectory computation ($40$ ?). Could you comment on the impact of truncated iterations on solution quality and computational efficiency for example if we divide by a factor of $4$ the number of iterations per time-step ?

– In the atmospheric context, Ritchie et al. (1995) report only a $20\%$ overhead of their SL version of the ECMWF model compared to the Eulerian version for a time-step multiplied by $5$. Do you think it is conceivable in the future to target such a small overhead within the NEMO framework ? The main difference probably comes from the fact that the computational cost of an atmospheric model is largely dominated by physical parameterizations. In NEMO, parameterizations only account for about $10\%$ of the cost while advection in the Eulerian case represents about $12\%$ on average ($8\%$ for tracer advection + $4\%$ for momentum advection). I think it makes it much more challenging in the oceanic context to deliver an SL implementation more efficient than Eulerian approaches.

– The parallel implementation of SL methods can be quite cumbersome. Is there a limitation on how far you go to find the departure points, do you

restrict the search for the departure points to the neighbouring MPI subdomains or can you go further ? Loop vectorization is important but with an SL approach you potentially also increase significantly the number of MPI exchanges especially with your iterative algorithm to find departure points ? Please also comment on your expectations in terms of scalability. My intuition is that the smaller the MPI subdomains the larger the computational overhead will be compared to an Eulerian approach.

**Minor comments :**

- **p. 1, l. 16-17:** What do you mean by "persistence of the initial conditions" ? Shouldn't it be "persistence of the boundary conditions" ?

- **p. 1, l. 18:** "systems" is repeated twice

- **p. 2, l. 49:** "CFL numbers" → "Courant numbers" (the CFL number is virtually infinite for a SL algorithm)

- **p. 4, Eq. (1):** The vector $x$ is not defined

- **p. 5, Eq. (3):** Is it really $x(t)$ ? Shouldn't it be $x(t_0)$.

- **p. 5, Eq. (4):** $f$ should not be in bold ?

- **p. 13, l. 312:** "CFL numbers" → "maximum Courant numbers"

- **p. 14, l. 327:** "is only stable to a CFL number of" → "is only stable for Courant numbers such that"

- **p. 14, l. 328:** "a CFL number of" → "for a maximum Courant number of"
[Figure]

- **l. 336,340,341,caption of Fig. 3,505,506,515,Fig. 5,552,581:** inconsistent use of CFL where it should refer to the maximum Courant number encountered in the simulation.

- **p. 22, l. 525:** This is a long-standing belief in the NEMO community but there is no such thing as a Total Variation Diminishing scheme in NEMO. The TVD acronym in the so-called TVD scheme in NEMO actually means *Tracer Variance Dissipation* and not *Total Variation Diminishing* which are completely different properties. The tracer advection scheme referred to as TVD scheme in NEMO is an FCT (Flux Corrected Transport) scheme, see (Lévy et al., 2001) for a description in the NEMO framework.

- **p. 22, footnote 9:** in the context of NEMO, the exact reference for this fix and for the consequences of not using it is Ducousso et al. (2017).

- **p. 24, l. 566:** "mangitude" → "magnitude"

*Florian Lemarié*

**References**

Ducousso, N., Sommer, J.L., Molines, J.M., Bell, M., 2017. Impact of the "symmetric instability of the computational kind" at mesoscale- and submesoscale-permitting resolutions. Ocean Modell. 120, 18 – 26.

Lauritzen, P.H., 2007. A stability analysis of finite-volume advection schemes permitting long time steps. Mon. Weather Rev. 135(7), 2658–2673.

Lemarié, F., Debreu, L., Madec, G., Demange, J., Molines, J., Honnorat, M., 2015. Stability constraints for oceanic numerical models: implications for the formulation of time and space discretizations. Ocean Modell. 92, 124 – 148.

Lévy, M., Estublier, A., Madec, G., 2001. Choice of an advection scheme for biogeochemical models. Geophys. Res. Lett. 28(19), 3725–3728.

Ritchie, H., Temperton, C., Simmons, A., Hortal, M., Davies, T., Dent, D., Hamrud, M., 1995. Implementation of the semi-lagrangian method in a high-resolution version of the ecmwf forecast model. Mon. Weather Rev. 123(2), 489–514.

Shchepetkin, A.F., 2015. An adaptive, courant-number-dependent implicit scheme for vertical advection in oceanic modeling. Ocean Modell. 91, 38 – 69.

Shchepetkin, A.F., McWilliams, J.C., 2005. The regional oceanic modeling system (roms): a split-explicit, free-surface, topography-following-coordinate oceanic model. Ocean Modell. 9(4), 347 – 404.

---

## Referee Comment (RC2) · Mike Bell (Referee) · 13 May 2020

**A Review of "Semi-Lagrangian advection in the NEMO ocean model"**

**By Christopher Subich, Pierre Pellerin, Gregory Smith, and Frederic Dupont**

This paper describes the implementation of a semi-Lagrangian advection scheme for momentum and tracers in the NEMO ocean model. The algorithms that have been adapted or developed are well presented and clearly explained in useful detail. Some preliminary results showing that the methods "work" are also presented for idealised and real-world configurations.

The adaptation for ocean models of semi-Lagrangian algorithms is a significant contribution to the ocean modelling literature and the paper deserves to be published for that reason alone. Similarly to the other reviewer, my view is that the implementation is a major achievement for which the authors deserve congratulation. I am also very supportive of the algorithms being honestly presented, warts and all. The initial results on the quality of the simulations are also interesting.

The first reviewer has produced a very thorough review so my review is less lengthy than it might have been. I refer to the first review where it seems relevant.

From my point of view the main weakness of the paper is in the introductory discussion. The motivation for using a semi-Lagrangian advection scheme needs more discussion. As the first reviewer mentions, Lemarie et al (2015) and Shchepetkin (2015) show that the time step in NEMO-like ocean models of ORCA025 resolution can be limited by vertical advection in hot-spots near the bathymetry (those papers should be referenced). As the authors mention ice-ocean drag can also limit the time-step. In the ocean, horizontal advection by internal gravity waves and currents is usually considered to be the next limitation on the time-step. In atmospheric models, semi-Lagrangian schemes usually calculate the horizontal pressure gradients as well as the Lagrangian advection semi-implicitly (Staniforth & Cote 1999) so that the time-step is not limited by the horizontal currents or the gravity wave speeds. The rationale for this is that external gravity waves (and sound waves) in the atmosphere travel at about 300 m/s whilst the winds themselves are typically around 100 m/s. In the ocean, internal gravity waves travel at about 3 m/s whilst ocean currents are not much more than 1.5 m/s. So one might anticipate that semi-Lagrangian schemes for the ocean would similarly calculate the (baroclinic) pressure gradients semi-implicitly. Unfortunately that would necessitate inversion of a 3D Helmholz problem. This problem would be much better conditioned than the free surface solver for the external mode but its efficient solution would be technically challenging. The second paragraph of section 1.1 highlights the point that the horizontal grid spacing in ORCA025 in the Canadian Archipelago is only 3-4 km.  On first reading I did wonder whether the internal gravity waves may be significantly slower in the regions where the grid spacing is small and whether that would make semi-Lagrangian advection efficient for this grid. But the authors do not mention that possibility.

I think there needs to be some recognition and discussion of the points made in the previous paragraph. It would be best for this to be in the introduction if good reasons can be given as to why semi-Lagrangian advection without semi-implicit calculation of pressure gradients is expected to allow long timesteps. Otherwise the issue should be, perhaps briefly, discussed in the concluding section.

The introduction should also mention the concerns that a) SL schemes usually do not conserve quantities in the same way that flux formulations do and b) their upstream nature introduces a biharmonic damping (see the suggestion relating to Figure 8 below).

**More detailed comments**

**Title:**

I agree with the other reviewer that there could usefully be a few more words in the title. Perhaps implementation [or development] of a Semi-Lagrangian algorithm [or scheme] …

**Abstract**

I am uncomfortable with the first paragraph of the abstract because it gives the impression that the semi-Lagrangian scheme described in this paper is very similar to the SISL algorithms used in atmospheric models which, for the reasons explained earlier, does not seem to me to be really the case.

**Introduction**

Lines 15-27: I know it is difficult to start papers but the discussion of coupled models here seems to be rather tangential.

Lines 29-39. The pole problem is of course much more severe in atmospheric models. Are the currents in these narrow straits as strong as those in the Gulf Stream? I ask because tidal currents can be very strong and CFL depends on u/dx. Is the water shallow in the Canadian archipelago where the grid spacing is smallest? The currents might then be much faster than the internal gravity waves in these regions (see earlier comment). This would make the SL method suitable for the ORCA025 domain – but not for some other domains.

Lines 40-43. The references given by the other reviewer should be mentioned here.

**Section 2**

The other reviewer asks questions about the details of the leapfrog scheme. NEMO is in the process of transitioning to an RK3 time-stepping scheme. Perhaps it is beyond the scope of the paper to discuss this but it might be easier (and more forward looking) to consider how to adapt the SL scheme for an RK3 scheme.

Equation (3) and line 95: I think x(t) should be x(t_0) (three occurrences two in equation (3)).

I am inclined to agree with the other reviewer that extending the notation in equations (1) – (6) would be helpful. Some ocean model readers might find a schematic figure helpful too. Equations (1)-(4) read very easily but on first reading it took me some time to understand that the first identity in (5) is just (1) with x written as x(t_0 + \Delta t). This was partly because I could not see what the comment on the line before (5) meant. That would be clear if it was written as "noting that x in (1) corresponds to x(t_0 + \Delta t) and that the evolution in (4) is due solely to advection so that RHS_f becomes RHS_{f,adv}, gives"

**Section 3**

Algorithm 1 and its description are very neat. Algorithm 2 is also clearly described. In the description of case 1 it is (9) rather than only (9a) that applies. If these algorithms have been presented in previous papers references to the original papers should probably be given.

Lines 199-202: I found it difficult to understand this paragraph and hence confusing on first reading. On second reading I think it does not say very much.

**Section 3.2**

Line 215: I was puzzled on first reading that the derivation for horizontal interpolation previously derived was discontinuous. Is the point simply that the derivatives are continuous in grid point space but not in physical space? It would be less disconcerting and easier to read if this point were made more directly.

Lines 262-265: "Imagine …" I found these two sentences difficult to follow. Is the point that the vertical grid locations vary in the horizontal because of the partial cells so that vertical interpolation to departure points is necessary even though w=0?

**Section 3.3**

The solution (12)-(14) of (11) is rather neat. Is this a standard test case in the semi-Lagrangian community? A reference for it would be appropriate.

**Section 4**

Lines 358-373 give a very helpful simple example motivating the solution proposed. The short paragraph in lines 374-376 is less clear. What follows is again very clear and helpful.

The calculation in lines 390 – 406 is admirably clear and concise.

**Section 4.3**

The use of corner solutions to supplement bi-linear interpolation looks quite novel to me. I'm impressed by it.

**Section 5.1**

Lines 483-484: This is a useful test case, but one which allows internal gravity waves to propagate faster than the advective velocities would be more relevant to most of the ocean.

Line 514 and Figure 5: "there is an inconsistency which grows with \delta t". I think the authors are pointing out that in (3) the terms are "centred" about the middle of the trajectory (half way between $x_a$ and $x_d$) whereas in (5) and (6) only the Lagrangian advection term is centred at that point, the $RHSE_f$ term being calculated solely at $x_a$. I think this point should be made near the end of section 2. Does it affect the order of accuracy of the scheme?

Footnote 9: Ducousso et al (2017) supports this statement

Lines 550-554: I hope the authors will sort out the ice-ocean coupling before re-submitting the paper so that integrations using longer time-steps are presented in the revised manuscript.

Line 557 and Figure 6: The MOC appears to be weaker in the SL integrations. The differences are at least comparable with those typically obtained between 1°, 1/4° and 1/12° simulations. As this is only one integration it is difficult to be sure how significant the difference is. But more should be said than just that they are comparable. The ACC appears to be stronger in the SL simulations. Similar variations can be obtained for ORCA025 with Eulerian schemes by adjustment of model parameters (though I'm not sure whether this is documented in the literature).

Lines 559-562: Much larger variations than those between the curves in Figure 7 are obtained by varying the model resolution from 1° to 1/4° to 1/12°

Lines 563-567 and Figure 8: NEMO is used extensively for climate change simulations. The implied change in ocean heat content could be significant. Erosion of the thermocline by numerical mixing is a major issue in ocean simulations for climate. So the authors should plot the global mean vertical

temperature profile at the start and end of the integration and compare with results with other papers. See for example Adcroft et al https://agupubs.onlinelibrary.wiley.com/doi/full/10.1029/2019MS001726 and Megann https://doi.org/10.1016/j.ocemod.2017.11.001

**Section 5**

Lines 577-579: A slightly longer summary of what has been achieved in the methods section would probably be helpful to the reader. New algorithms and arguments should be highlighted here or in the introduction.

Lines 590-593: The authors should mention the need to increase the halo size for communication between processors and the possibility of load imbalances when some iterations are slow to converge.

---

## Author Response (AR1)

**Authors' response to reviewers for "Semi-Lagrangian advection in the NEMO ocean model"**

June 26, 2020

To begin, we are thankful for the review comments provided by the reviewers Florian Lemarié (reviewer #1) and Mike Bell (reviewer #2). Collectively, the reviews have highlighted areas where the manuscript was lacking, particularly with respect to its clarity and presentation. We have revised the manuscript with this feedback in mind, making changes to incorporate or otherwise address this feedback.

Because both reviewers were in broad agreement about the weaknesses of the paper and did not provide conflicting feedback, a summary of changes is listed below by section. Additionally, the revised manuscript and a "diff" of changes will be submitted after this comment is finalized.

**Title and abstract**

Both reviewers noted that the original title of the article was too bold, to put it bluntly. Additionally, the executive editor pointed out the GMD requirement that papers which refer to development for a single model must mention the model name and version number in the title. Consequently, the title is now "Development of a semi-Lagrangian advection scheme for the NEMO ocean model (3.1)."

Reviewer #2 additionally noted that the original abstract gave the impression that the method developed in this article is "very similar to the SISL algorithms used in atmospheric models." The first paragraph of the abstract is now revised to hopefully make the distinction more clear, and the second paragraph is reworded for simplicity.

**Introduction**

The introduction has been greatly expanded in response to the reviewers' comments. In textual order:

- Reviewer #2 noted that the leading paragraphs on coupled modeling seemed to be a tangential motivation. This has the awkward characteristic of being tangential yet true – the idea of applying semi-Lagrangian advection to the ocean model at CMC came out of realizing the computational cost of running coupled forecast systems. This section has been revised and modestly expanded to make the practical focus more clear.

- Both reviewers note that the discussion of why semi-Lagrangian advection might help the timestep size in ocean models was lacking. This is now more comprehensibly discussed in new section 1.1, which draws the suggested direct contrast between timestep-limiting factors in atmosphere and ocean circulation models. Grid stretching is now a subsection to this discussion, which (at Reviewer #2's request) now also includes a brief description of the ocean flows in the gridpoint-clustered portion of the Canadian Arctic Archipelago.

- The "existing work" subsection (now 1.2) more directly engages with the literature on conservation-preserving semi-Lagrangian methods (noting this as not implemented but a future possibility, and without this interpretation semi-Lagrangian advection has a finite-difference interpretation) and ALE coordinates.

The other points raised by reviewer #1 (the "two other levels of constraints") are generally agreed to but addressed in the main body of the text as the issues arise.

**Time discretization**

- Both reviewers note that the notation in this section was awkward. Consequently, we have entirely revised this section to use a simpler notation of $f^B$ ("before"), $f^N$ ("now"), and $f^A$ ("after") that should be familiar to readers from the leapfrog context, adapting it to semi-Lagrangian advection to add $f^D$ ("departure"). This also resulted in small changes to the notation in subsequent sections for consistency.

- This section is also reorganized to separate the semi-Lagrangian advection (2.1) from its reconciliation with the leapfrog algorithm (2.2) to clarify the changes in perspective.

- Reviewer #1 also expressed doubts about whether semi-Lagrangian advection as-defined was robust to the Asselin timestepping filter. This analysis is now present in the new subsection (2.3), and the Asselin filter does not negatively affect the stability of semi-Lagrangian advection as-implemented.

**Interpolation**

- Reviewer #2 remarked that the discussion of two-dimensional interpolation was cumbersome and verbose. What was formerly the non-numbered "two-dimensional application" subsection has been removed, with the comment briefly summarized and placed just before the slope-limiting discussion.

- Reviewer #2 also noted that the discussion of vertical advection was confusing, especially the claims about discontinuous derivatives. Subsection 3.2 has been revised and reworded.

- We also revised the subsequent discussion of vertical slope-limiting to clarify (at reviewer #2's note of confusing language) why it is necessary at the bottom boundary in the presence of partial cells.

- A reference (Turkington et al., 1991) has been added for the numerical example on this section, at reviewer #2's request. To our knowledge this is not a standard test-case in the semi-Lagrangian literature, but nonlinear generalizations of this approach are a standard technique for calculating the profiles of nonlinear internal gravity waves.

- Reviewer #1 noted that the description of the advection constraint for the Eulerian/leapfrog numerical example in this section was "fuzzy," and so we have adopted the more precise definition. This led to no practical difference in the calculation, since the maximum Courant number in the domain is reached at the top and bottom boundaries where the vertical velocity is zero. (This did, however, lead to a discovery of a small bug in the code that generated this figure, which used the wrong vertical mode number to calculate wave-induced horizontal velocities for the purposes of evaluating the Courant number. This has been addressed in the submitted code repository and the figure regenerated; there is essentially no difference in results.)

  Additionally, we replaced "CFL number" in the paper with "Courant number" throughout, since the latter concept is indeed the intended use of the term.

- Reviewer #1 also inquired about the performance of the Eulerian/leapfrog method in this section with a maximum Courant number close to 1. We investigated this over the range 0.2–0.99 and found little difference in error compared to the exact solution; this is mentioned in-text rather than by adding more lines to figure 3.

**Trajectory calculation**

- Reviewer #2 notes that the discussion about extrapolating into the boundary is confusing, and reviewer #1 asks whether this semi-Lagrangian method faces a Lipschitz stability condition. These are the same issue: the problem of extrapolating into the boundary arises only when a calculated trajectory would cross that of a fluid parcel that begins and remains on the (no normal flow) boundary. Consequently, we have revised the first part of subsection 4.1 to make this connection.

**Numerical results**

- Reviewer #2 requested more clarification on the inconsistency between the semi-Lagrangian advection and the Eulerian application of forcing. This amounts to an $O(\Delta t)$ approximation in the integral form of the semi-Lagrangian advection equation, and this is now noted in the discussion in subsection 5.1.

- Reviewer #1 notes that NEMO's TVD scheme is really a "tracer variance dissipation" scheme. This has been changed throughout with a citation to Lévy et al. (2001) at the first mention.

- Both reviewers remarked on the relatively short timestep used in the ORCA025 runs of section 5.2. New footnote 8 has been added to provide more context; in brief the ice/ocean drag parameter is increased following Roy et al. (2015), which makes the problem more apparent for the operational forecasting configuration than for typically-presented runs. At the same time, we wanted to maintain the same physical parameterizations between the operational configuration and the runs presented in this paper. Addressing this problem would be ideal and is the focus of ongoing work, but the runs of section 5.2 took long enough to complete on the shared supercomputing resources that they cannot be practically be repeated in GMD's peer-review timeframe even if a solution were immediately at hand.

- Reviewer #1 asked about the number of iterations taken to find trajectories and the effect of trajectory truncation. This is now discussed further in section 5.2; the mean number of trajectory iterations per cell for the semi-Lagrangian tracer run was 1.004, so truncated trajectories were truly exceptional. The performance cost of trajectory iteration is also addressed in the conclusions.

- Reviewer #2 requested expanded commentary on the MOC and circumpolar current results, which we have provided. Because these runs do report preliminary results, we want to be cautious about reporting false confidence that semi-Lagrangian advection causes physically-relevant changes in results that may not in fact be robust, but we agree that we erred on the terse side here.

- Reviewer #2 also requested a look at the mean global temperature profile at the start and end of the simulation. This is the new figure 9, with brief discussion at the end of section 5.

**Conclusions**

- Reviewer #2 requested a longer summary of the achievements, particularly one that highlights new algorithms. This is now added at the beginning of section 6, where we have added a list that highlights the core algorithms of this paper.

- Both reviewers had questions about the performance of the method and its parallel implementation. This is now dealt with in the conclusions, under the new (non-numbered) "performance and implementation" subsection.

- Reviewer #2 requested a deeper look at the application to climate simulations, and consequently we have expanded the discussion in the commentary on the results. The temperature profile results (specifically temperature stability in deep water) seem to be encouraging for climate applications, but we reserve a full recommendation for a future day when either temperature/salinity drift is fully characterized (and found to be acceptable) or conservation is explicitly added.

- Reviewer #2 also requested a brief discussion of how this algorithm might apply to the RK3 timestepping algorithm used in upcoming versions of NEMO. Since this is very much "future work" for both NEMO and semi-Lagrangian advection, we have added this discussion to the conclusions.

**References**

[revised manuscript text omitted]
\,A}(x_{ref}) = f^{t_0-\Delta t\,B}(x_{ref}) + 2\Delta t \cdot \mathrm{RHSE}_f \mathbf{F}(x_{ref}), \tag{1}$$

where $\mathrm{RHSE}_f$ (Eulerian right-hand side) generally stands in for both true forcing terms (such as heating or evaporation) as well as time-tendency terms from the transport and momentum equations. For advective processes $f^A$ is the field calculated at time
140 $t_0 + \Delta t$, $f^B$ is the field evaluated at the known prior time $t_0 - \Delta t$ ("before"), $f^N$ is the field at the provided time $t_0$ ("now"), and $\mathbf{F}$ is the forcing operator. The forcing operator includes advective processes at the RHSE terms arising from tracer and momentum flux are evaluated at the current "now" time-level, whereas RHSE terms arising from but diffusive, damping, and hydrostatic pressure may terms might be evaluated at either the previous or new "before" or "after" time-levels.

This is an Eulerian approach to fluid motion, where tracer and momentum values are tracked at specific locations (namely
145 the grid points) over time along the fixed reference grid at all times, and fluid flows through this grid.

**2.1 Semi-Lagrangian advection**

In contrast, the semi-Lagrangian advection scheme considers Lagrangian advection schemes consider the fluid parcel to be the fundamental unit of discretization. In this perspective, if $f$ is a property of a fluid parcel that is conserved along a trajectory[1],
* * *
[1]This is true for temperature, salinity, and momentum provided the ocean is treated as an incompressible fluid. This assumption is satisfied by NEMO-OPA's adoption of the Boussinesq approximation.

it satisfies the continuous equations:

$$\frac{D}{Dt}f(\boldsymbol{x}(t)) = \text{RHSL}_f \mathbf{F}_L(\boldsymbol{x}(t)), \tag{2}$$

where $\frac{D}{Dt} = \partial_t + \boldsymbol{u} \cdot \nabla$ is the material derivative and  $\mathbf{F}_L$ (Lagrangian right-hand side) contains all the same forcing terms as  $\mathbf{F}$ *except* those arising from tracer and momentum flux, which are included inside the material derivative.

Ordinarily, (2) is discretized so that  $\mathbf{F}_L$ is evaluated following the Lagrangian particles  in the moving coordinate frame $\boldsymbol{x}(t)$, satisfying the trajectory equation:

$$\frac{D}{Dt}\boldsymbol{x}(t) = \boldsymbol{u}(\boldsymbol{x}(t)). \tag{3}$$

From an Eulerian point of view, (3) is a trivial identity based on the definition of the material derivative, but from the Lagrangian point of view (3) must be solved to define $\boldsymbol{x}$ over time.

One technique for solving (2) and (3) is the two time-level implicit semi-Lagrangian method, used in the GEM atmospheric model (Girard et al., 2014) among others. Here, the $\mathbf{F}_L$ terms are evaluated  with a trapezoidal rule

$$f^{t_0+\Delta t}(\boldsymbol{x}(t_0 + \Delta t)) = f^{t_0}(\boldsymbol{x}(t)) + \frac{\Delta t}{2}\left(\text{RHSL}_f^{t_0+\Delta t}(\boldsymbol{x}(t_0 + \Delta t)) + \text{RHSL}_f^{t_0}(\boldsymbol{x}(t))\right),$$

, discretizing (2) and (3) as:

$$f^A(\boldsymbol{x}_{ref}) = f^N(\boldsymbol{x}^D) + \frac{\Delta t}{2}\left(\mathbf{F}_L^A(\boldsymbol{x}_{ref}) + \mathbf{F}_L^N(\boldsymbol{x}^D)\right) \text{ and} \tag{4a}$$

$$\boldsymbol{x}_{ref} = \boldsymbol{x}^D + \frac{\Delta t}{2}\left(\boldsymbol{u}^A(\boldsymbol{x}_{ref}) + \boldsymbol{u}^N(\boldsymbol{x}^D)\right). \tag{4b}$$

The trajectory equation (4b) acts to implicitly define the paths of the traced fluid parcels, where each location on $\boldsymbol{x}_{ref}$ is associated with a corresponding departure-point location $\boldsymbol{x}^D$. Over the single timestep, fluid parcels depart from $\boldsymbol{x}^D$ (which in general is not aligned with the grid) and arrive on the reference grid.

The forcing This off-grid, departure point evaluation of $\boldsymbol{u}$ and $\mathbf{F}_L$ is fundamental to Lagrangian and semi-Lagrangian methods, and $f^N(\boldsymbol{x}^D)$ ($\mathbf{F}_L^N(\boldsymbol{x}^D)$) can be written more simply as $f^D$ ($\mathbf{F}_L^D$) for "departure-point $f$ ($\mathbf{F}$)." Neither the time-implicit evaluations (generally) nor the off-grid evaluations (of non-advective forcing) are compatible with the core structure of NEMO-OPA, which considers advection to be just one of many  independent operators influencing the $\mathbf{F}$ term of (1).

**2.2 Reconciliation**

Implementing semi-Lagrangian advection in NEMO-OPA requires adopting as much of the framework of (1) as possible, without changing the evaluation of non-advective forcing terms. Effectively, the semi-Lagrangian advection routine must ultimately supply a time-trend that, from the perspective of the leapfrog timestep algorithm, is indistinguishable from a conventional flux-form advection operator.

To effect this, consider (2) without forcing terms ($\mathbf{F}_L = 0$) over the interval $t_0 - \Delta t$ to $t_0 + \Delta t$. The function $f$ is preserved following the flow, so this gives the simply-written:

$$f^{t_0+\Delta t}((t_0 + \Delta t)) = f^{t_0-\Delta t}((t_0 - \Delta t)). \tag{5}$$

This is approximated by taking one timestep of (1) (with only advective forcing $\mathbf{F}_{adv}$), but the latter involves integrating over the whole interval from $t_0 - \Delta t$ to $t_0 + \Delta t$. Thus, we should identify $f^D$ (and the departure points generally) not with the "now" time-level in the leapfrog scheme, but with the "before" time-level. Doing so and equating (5) and (1) gives:

$$f^{t_0+\Delta t}((t_0 + \Delta t)) = f^{t_0-\Delta t}((t_0 + \Delta t)) + 2\Delta t\cdot\mathbf{F}_{adv} = f^{t_0-\Delta t}(x(t_0 - \Delta t)), \
[revised manuscript text omitted]

---

## Referee Report (RR1)

**A Review of**
**"Development of a semi-Lagrangian advection scheme for the NEMO ocean model (3.1)"* (Revision 1)**
**by C. Subich, P. Pellerin, G. Smith, and F. Dupont**
* * *
I am satisfied with the modifications brought to the manuscript and I thank the authors for answering with care to my remarks. Besides the minor comments given below, I consider the manuscript ready for publication.

**Minor comments :**

- **p. 2, l. 27-28:** "Courant-Frederichs-Lwey" → "Courant-Friedrichs-Lewy"
- **p. 2, l. 29-31:** while the Euler equations are indeed an hyperbolic system, a consequence of the hydrostatic assumption is however that the primitive equations (solved by NEMO) are no longer hyperbolic. It is just that for your study you consider only the hyperbolic part of the primitive equations system.
- **p. 2, l. 38:** it is not the hydrostatic assumption that eliminate sound waves but the Boussinesq assumption.
- **p. 5, l. 97-99:** I don't think the discussion about the z-tilde ($\tilde{z}$) coordinate is essential for your paper. I would suggest to remove the paragraph starting l. 95. I make this suggestion mostly because I don't understand your remark about the fact that the $\tilde{z}$-coordinate will increase the maximum Courant number in the vertical. From my understanding it is the other way around, because of the ALE treatment, part of the vertical velocity will be treated in a Lagrangian way and thus won't be subject to stability criteria. The residual will be treated in an Eulerian way but this residual is expected to be smaller than the full vertical velocity handled with a quasi-Eulerian coordinate like $z^{\star}$.

---

## Author Response (AR2)

**Authors' response to reviewers for "Semi-Lagrangian advection in the NEMO ocean model"**

July 30, 2020

Once again, we are thankful for the reviewer comments. We have made the suggested changes:

- The typo of "Courant-Friedrichs-Lewy" on page 2 has been corrected.

- We no longer call the hydrostatic equations a hyperbolic system on page 2.

- We now note that the Boussinesq assumption eliminates sound waves, also on page 2.

- We have clarified the language regarding the $\tilde{z}$ coordinate on page 5. As the reviewer suggests, we did intend to say that the $\tilde{z}$ coordinate improves model stability with respect to vertical motion, but the language was unclear.

  The reviewer was right to note in the first set of comments (comments of Florian Lemarie, page C4) that this coordinate system is becoming more popular, and it is appropriate to give due credit to the existing literature.

The annotated manuscript is attached. Changes from the initial revision are confined to pages 2 and 5.

[revised manuscript text omitted]